# Graph-Structured Gaussian Processes for Transferable Graph Learning

**Jun Wu[1], Elizabeth Ainsworth[1,2], Andrew Leakey[1], Haixun Wang[3], Jingrui He[1]**
[1]University of Illinois at Urbana-Champaign
[2]USDA ARS Global Change and Photosynthesis Research Unit
[3]Instacart
{junwu3,ainswort,leakey,jingrui}@illinois.edu, {haixun}@gmail.com

## Abstract

Transferable graph learning involves knowledge transferability from a source graph to a relevant target graph. The major challenge of transferable graph learning is the distribution shift between source and target graphs induced by individual node attributes and complex graph structures. To solve this problem, in this paper, we propose a generic graph-structured Gaussian process framework (`GraphGP`) for adaptively transferring knowledge across graphs with either homophily or heterophily assumptions. Specifically, `GraphGP` is derived from a novel graph structure-aware neural network in the limit on the layer width. The generalization analysis of `GraphGP` explicitly investigates the connection between knowledge transferability and graph domain similarity. Extensive experiments on several transferable graph learning benchmarks demonstrate the efficacy of `GraphGP` over state-of-the-art Gaussian process baselines.

## 1 Introduction

Transfer learning [40, 54] aims at studying the transfer of knowledge or information from a source domain to a relevant target domain under distribution shifts. The knowledge transferability across domains has been investigated from various aspects. For example, [3, 15, 67] proposed to empirically estimate the data-level domain discrepancy for measuring the transferability across graphs. [8, 49] adopted adaptive Gaussian processes with transfer kernels, which formulated the task relatedness as a hyper-parameter for tuning. [22, 36, 63] instead evaluated the transferability of a pre-trained source model in the target domain. However, most existing works followed the IID assumption that samples are independent and identically distributed

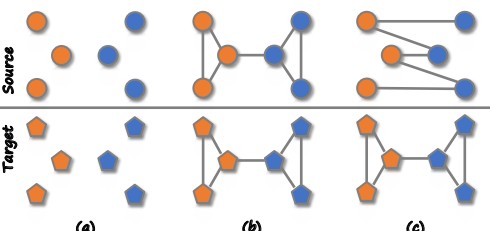

Figure 1: Illustration of transferable graph learning. (a) Samples are independently drawn. (b) Samples are connected in each domain, and both domains follow the homophily assumption. (c) Source and target domains follow different graph structure assumptions.

in each domain. This might limit their capacities in understanding the knowledge transferability across domains with non-IID data, e.g., cross-domain recommendation [55], cross-network role identification [69], etc.

In this paper, we focus on the problem of transferable graph learning over non-IID graph data. The challenge of transferable graph learning induced by the graph structure information can be explained in Figure 1. It can be seen that Figure 1(a) provides source and target samples associated with input features and output class labels (indicated by node colors). Suppose that source and target

distributions have been aligned (e.g., optimized by domain discrepancy minimization algorithms [15, 67]), Figure 1(a) implies that source and target domains have similar data distributions in the feature space, thereby enabling feasible knowledge transfer across domains. Compared to Figure 1(a), Figure 1(b)(c) introduce additional graph structure information. In Figure 1(b), source and target graphs follow similar structures where nodes within the same class tend to be connected (i.e., the homophily assumption [59]). In contrast, the structures of source and target graphs fundamentally differ in Figure 1(c). That is, though the target graph follows the homophily assumption, the source graph holds the heterophily assumption that connected nodes may have different classes and dissimilar node attributes [68]. Therefore, compared to standard transfer learning [31, 40] with IID data, transferable graph learning is much more challenging because knowledge transferability across graphs can be determined by both individual node features and complex graph structures.

In recent years, transferable graph learning has been studied [55, 58, 66, 69] where both source and target graphs follow the homophily assumption (Figure 1(b)). However, little effort has been devoted to investigating the knowledge transferability between homophilic and heterophilic graphs (Figure 1(c)). To bridge this gap, in this paper, we focus on studying the knowledge transferability across graphs with either the same (e.g., homophily or heterophily) or different assumptions.

We start by introducing a universal structure-aware neural network to build the input-output relationship between structure-aware input sample $(v, G)$ and its associated output value $y_v$ for tackling homophilic and heterophilic graphs. This model encodes both local node representation (sample-level) and global graph representation (domain-level) simultaneously. The crucial idea is to automatically select relevant neighborhoods by adding learnable weights to neighborhood aggregation. Thus, it enables knowledge transferability between homophilic and heterophilic graphs by adaptively selecting neighborhoods for each graph. The intuition is that source and target graphs might have common knowledge from the selected neighborhoods, e.g., nearby neighborhoods for the homophilic graph and high-order neighborhoods for the heterophilic graph. Then, we show the equivalence between structure-aware neural network and graph Gaussian process in the limit on the layer width. This observation sheds light on building graph-structured Gaussian processes (GraphGP) for transferable graph learning. Moreover, the generalization analysis of GraphGP shows the connection between knowledge transferability and graph domain similarity. Compared to previous works [45, 58, 69], GraphGP benefits from (i) feasible knowledge transferability between homophily and heterophily graphs, and (ii) flexible incorporation with existing graph neural networks and graph kernels. Extensive experiments on node regression tasks demonstrate the effectiveness of GraphGP over state-of-the-art Gaussian process baselines. The main contributions of this paper are summarized as follows.

- A generic graph-structured Gaussian process framework GraphGP is proposed for transferable graph learning. It is derived from a structure-aware neural network encoding local node representation (sample-level) and global graph representation (domain-level) simultaneously.

- We show that GraphGP tackles the knowledge transferability in homophily and heterophily graphs using a simple neighborhood selection strategy. In addition, we theoretically analyze the knowledge transferability across graphs from the perspective of graph domain similarity.

- Experimental results on a variety of graph learning benchmarks demonstrate the efficacy of GraphGP over state-of-the-art Gaussian process baselines.

The rest of this paper is organized as follows. Section 2 summarizes the related work, and Section 3 provides the problem definition of transferable graph learning. In Section 4, we present the graph-structured Gaussian process (GraphGP) for transferable graph learning. The experimental results are provided in Section 5. Finally, we conclude the paper in Section 6.

## 2 Related Work

**Transfer Learning:** Transfer learning [40, 54] tackles the transfer of knowledge or information from a source domain to a relevant target domain. It is theoretically shown [3, 67] that the generalization performance of a learning algorithm can be improved by leveraging latent knowledge from the source domain. Specifically, the generalization error of a learning algorithm on the target domain can be bounded in terms of the source knowledge and the distribution discrepancy across domains. Recently, transfer learning has been explored in non-IID graph data [7, 55, 58, 64, 69]. However, most existing works focus on either investigating the transferability of a pre-trained graph neural

network [27, 45, 69] or designing graph transfer learning algorithms [58, 66] with relaxed assumptions (e.g., $k$-hop ego-graphs are assumed to be independent and identically distributed [69]).

**Gaussian Process:** Gaussian process [9, 42] provides a principled nonparametric framework for learning stochastic functions from observations. It has been applied to transfer learning when samples in each domain are independent and identically distributed [8, 49, 56]. The crucial idea is to infer the task relatedness between source and target domains using available labeled source and target examples. More recently, Gaussian processes have also been investigated in graph learning tasks, e.g., link prediction [14, 38, 65] and node regression [6, 21, 28, 35, 37, 39]. However, little effort has been devoted to understanding transferable graph learning from Gaussian process perspectives.

## 3 Preliminaries

### 3.1 Notation and Problem Definition

Suppose that a graph is represented as $G = (V, E)$, where $V = \{v_1, \cdots, v_{|V|}\}$ is the set of $n$ nodes and $E \subseteq V \times V$ is the edge set in the graph. Each node $v \in V$ can be associated with a $D$-dimensional feature vector $\mathbf{x}_v \in \mathbb{R}^D$. In the node regression tasks, the ground-truth output of each node is denoted by $y_v \in \mathbb{R}$. The structure of a graph $G$ can also be represented by an adjacency matrix $\mathbf{A} \in \mathbb{R}^{|V| \times |V|}$, where $\mathbf{A}_{ij}$ is the edge weight between $v_i$ and $v_j$ within the graph. Following [55, 69], we introduce a generic problem setting of transferable graph learning as follows.

**Definition 3.1** (*Transferable Graph Learning*). Given a source graph $G_s = (V_s, E_s)$ and a target graph $G_t = (V_t, E_t)$, *transferable graph learning* aims to improve the prediction performance of a graph learning algorithm in the target graph using latent knowledge from a relevant source graph.

### 3.2 Message-Passing Graph Neural Networks

Weisfeiler-Lehman (WL) graph subtree kernel [46] measures the similarity of two input graphs, inspired by the Weisfeiler-Lehman test of isomorphism [53]. The crucial component of WL subtree kernel is to recursively represent the subtree structure rooted at each node. This has motivated a variety of message-passing graph neural networks (GNN), which recursively learn the node presentation of node $v$ by aggregating the feature vectors from $v$'s local neighborhood [17, 18, 20, 57, 59]. Generally, a graph convolutional layer of message-passing GNNs can be summarized as follows.

$$h_v^{(l+1)} = \texttt{COMBINE}^{(l)} \Big( \underbrace{h_v^{(l)}}_{\text{Seed node representation}}, \underbrace{\texttt{AGGREGATE}_1^{(l)} \left( \{h_u^{(l)} | u \in N_1(v)\} \right)}_{\text{1-order neighborhood representation}}, \\ \cdots, \underbrace{\texttt{AGGREGATE}_k^{(l)} \left( \{h_u^{(l)} | u \in N_k(v)\} \right)}_{k\text{-order neighborhood representation}} \Big) \tag{1}$$

where $\texttt{COMBINE}^{(l)}$ function compresses the representations from the node itself and its neighbors at the $l^{\text{th}}$ graph convolutional layer into a single representation, and $\texttt{AGGREGATE}_j^{(l)}$ $(j = 1, \cdots, k)$ function aims to aggregate message from $j$-order neighborhood $N_j(v)$ of node $v$. It indicates that there are two major components for designing graph convolutional layers in GNNs: neighborhood selection and aggregation. More specifically, neighborhood aggregation allows compressing the graph structure and node attributes simultaneously, while neighborhood selection provides the flexibility of GNNs in tackling graphs with different assumptions (e.g., homophily [20] or heterophily [68]).

**Homophily graphs:** Homophily graph holds the assumption that nodes within the same class tend to be connected in the graph, e.g., citation networks [13], social networks [33], etc. This assumption has motivated various instantiations of message-passing graph neural networks over first-order neighborhoods, e.g., GCN [25], GraphSAGE [20], GAT [50], GIN [59].

**Heterophily graphs:** Heterophily graph holds that connected nodes may have different class labels and dissimilar node attributes, e.g., different classes of amino acids are more likely to connect within protein structures [30, 68]. Recently, by adaptively exploring the potential homophily in a high-order local neighborhood (e.g., $k > 1$), message-passing graph neural networks have been proposed for heterophily graphs, e.g., MixHop [1], H2GCN [68], GPR-GNN [11], HOG-GCN [52], GloGNN [29].

# 4 Methodology

In this section, we propose the graph-structured Gaussian processes for transferable graph learning.

## 4.1 Structure-Aware Neural Network

We start by showing the graph sampling process from a probability distribution space $\mathscr{P}$. A graph $G = (V, E)$ is sampled by first sampling a graph domain distribution $\mathbb{P}_\mathcal{G}$ from $\mathscr{P}$, and then sampling a specific graph $G$ from $\mathbb{P}_\mathcal{G}$. More specifically, one realization from $\mathscr{P}$ is a graph domain distribution $\mathbb{P}_\mathcal{G}$ (also denoted as $\mathbb{P}_\mathcal{G} = (\mathbb{P}_\mathcal{V}, \mathbb{P}_\mathcal{E})$ for node and edge sampling distribution) characterizing a two-stage graph sampling as follows. A set of nodes $V$ is sampled from the graph distribution $\mathbb{P}_\mathcal{G}$ (more specifically, sampled from $\mathbb{P}_\mathcal{V}$), and then the edge weight is sampled from $\mathbb{P}_\mathcal{G}$ (more specifically, sampled from $\mathbb{P}_\mathcal{E}$) over any pair of nodes in $V$. In addition to the graph structure (induced by the node dependence), the output label of a node can also be sampled accordingly in the context of node regression tasks. That is, a label-informed graph domain distribution $\mathbb{P}_{\mathcal{G},\mathcal{Y}}$ is sampled from $\mathscr{P}$. Next, a graph $G = (V, E)$ is sampled from $\mathbb{P}_\mathcal{G} = (\mathbb{P}_\mathcal{V}, \mathbb{P}_\mathcal{E})$, and then the output labels of nodes are sampled from $\mathbb{P}_{\mathcal{Y}|\mathcal{G}}$. The goal of node regression task [55, 69] is to learn a prediction function $f(\cdot)$ that predicts $\hat{y}_v = f(v, G)$ for each node $v \in V$.

In this paper, we consider the covariate shift assumption [40] over graphs. That is, the source and target graphs share conditional distribution $\mathbb{P}_{\mathcal{Y}|\mathcal{G}}(y|(v, G))$ but different marginal distributions $\mathbb{P}_\mathcal{G}(v, G)$. The aforementioned graph sampling process motivates us to learn the input-output relationship between an input sample $(G, v)$ and its associated output $y_v$ for transferable graph learning. To this end, for each input graph $G$ (e.g., source $G_s$ or target graph $G_t$ in transferable graph learning), we define a structure-aware neural network $f(\cdot, \cdot)$ as follows.

$$f_i^{(l)}(v, G) = \frac{1}{\sqrt{M}} \sum_{j=1}^{M} \mathbf{W}_{ij}^{(l)} \cdot \mu_j^{(l)}(v|G) \cdot \nu_j^{(l)}(G)$$

$$\mu^{(l)}(v|G) = h_v^{(l)} \quad \text{and} \quad \nu^{(l)}(G) = \texttt{READOUT}\left(\tilde{h}_v^{(l)}|v \in V\right)$$

(2)

where $M$ is the layer width[1]. $\mu^{(l)}(v|G)$ denotes the node (sample-level) representation of $v$ given the graph $G$ at the $l^{\text{th}}$ layer, and $\nu^{(l)}(G)$ denotes the graph (domain-level) representation of the graph $G$. Here READOUT function compresses a set of node representations into a single vector representation [59, 62]. This definition indicates that the embedding representation of the input $(v, G)$ (e.g., a pair of a node $v$ and its associated graph $G$) is given by two crucial components: an individual node embedding representation $\mu^{(l)}(v|G)$ and a global graph embedding representation $\nu^{(l)}(G)$. $h_v^{(l)}$ and $\tilde{h}_v^{(l)}$ represent graph neural networks to learn node and graph representations separately.

The intuition behind this definition is explained below. The graph $G$ is a realization of a graph domain distribution $\mathbb{P}_\mathcal{G}$. Then, the embedding representation of the graph $G$ can be explained as an empirical estimate of the distribution $\mathbb{P}_\mathcal{G}$, i.e., $\nu(G) \approx \nu(\mathbb{P}_\mathcal{G})$. Moreover, when the size of the graph $G$ goes to infinity (i.e., associated with an infinite number of nodes), $G$ approximates the true distribution $\mathbb{P}_\mathcal{G}$ and the embedding representation of the graph $G$ will be able to recover the true embedding of probability distribution $\mathbb{P}_\mathcal{G}$ [19, 48]. Besides, the embedding representation of a node $v$ can also be explained as an approximation of $\mathbb{P}_\mathcal{G}$ at the location $v$, i.e., $\mu(v|G) \approx \mu(v|\mathbb{P}_\mathcal{G})$.

**Remark.** The structure-aware neural network Eq. (2) is defined over message-passing graph neural networks commonly used for various single-domain graph learning tasks [18, 20, 23, 59]. Different from previous works, we focus on transferable graph learning scenarios across graphs. This motivates us to consider the task relatedness between source and target graphs, which can be explicitly measured by the domain distribution similarity [15, 56]. To this end, we design the structure-aware neural network Eq. (2) for learning domain-aware node presentation, i.e., the integration of the node (sample-level) representation and the entire graph (domain-level) representation. Specifically, we can show the connection between graph domain distribution and graph representation learning in the reproducing kernel Hilbert space for cross-graph learning tasks (see Corollary 4.4).

---

[1]Here, "layer width" indicates the number of neurons in a graph convolutional layer.

## 4.2 Graph-Structured Gaussian Process

Inspired by [11, 52, 68], we derive a universal message-passing graph neural network Eq. (1) for both homophily and heterophily graphs. More specifically, the message-passing graph convolutional layer of Eq. (1) for node $v$ within an input graph $G$ can be defined as follows.

$$\mathbf{h}_v^{(l)} = \sum_{i=0}^{k} \alpha_i \left( \frac{1}{\sqrt{M}} \sum_{u \in N_i(v)} \mathbf{W}_{\text{SANN}}^{(l)} \mathbf{x}_u^{(l)} + \mathbf{b}_{\text{SANN}}^{(l)} \right) \quad \text{and} \quad \mathbf{x}_u^{(l)} = \phi \left( \mathbf{h}_u^{(l-1)} \right) \tag{3}$$

where $\mathbf{x}_u^{(0)} = \mathbf{x}_u \in \mathbb{R}^D$, $N_0(v) = \{v\}$ denotes the seed node, $\mathbf{W}_{\text{SANN}}^{(l)}, \mathbf{b}_{\text{SANN}}^{(l)}$ are the weight and bias parameters, and $\alpha_i$ explicitly indicates the importance of the $i$-order neighborhood in finding relevant neighbors around node $v$. Similarly, we define the domain-level graph neural layer $\tilde{\mathbf{h}}_v^{(l)}$ parameterized by different $\tilde{\mathbf{W}}_{\text{SANN}}^{(l)}, \tilde{\mathbf{b}}_{\text{SANN}}^{(l)}$ and shared $\alpha_i$.

$$\tilde{\mathbf{h}}_v^{(l)} = \sum_{i=0}^{k} \alpha_i \left( \frac{1}{\sqrt{M}} \sum_{u \in N_i(v)} \tilde{\mathbf{W}}_{\text{SANN}}^{(l)} \tilde{\mathbf{x}}_u^{(l)} + \tilde{\mathbf{b}}_{\text{SANN}}^{(l)} \right) \quad \text{and} \quad \tilde{\mathbf{x}}_u^{(l)} = \phi \left( \tilde{\mathbf{h}}_u^{(l-1)} \right) \tag{4}$$

Generally, it is flexible to apply different model architectures to learn the node (sample-level $\mu^{(l)}(v|G)$) and graph (domain-level $\nu^{(l)}(G)$) representations. For simplicity, we adopt the same model architecture but different model parameters to learn node-level and graph-level representations. It is noteworthy that the neighborhood importance $\alpha_i$ is shared because both node and graph representation would share the same assumption (homophily or heterophily) for a given graph. Then, the structure-aware neural network Eq. (2) can be given by

$$f_i^{(l)}(v, G) = \frac{1}{\sqrt{M}} \sum_{j=1}^{M} \mathbf{W}_{ij}^{(l)} \cdot \mu_j^{(l)}(v|G) \cdot \nu_j^{(l)}(G)$$

$$\text{where} \quad \mu^{(l)}(v|G) = \mathbf{h}_v^{(l)} \quad \text{and} \quad \nu^{(l)}(G) = \frac{1}{|V|} \sum_{v \in V} \tilde{\mathbf{h}}_v^{(l)} \tag{5}$$

where the READOUT function of Eq. (2) is instantiated with mean pooling [62]. Different from previous works [11, 52, 68], we would take the neighborhood importance scores $\alpha_i$ as hyper-parameters for building transferable graph Gaussian process model as follows.

**Theorem 4.1.** *Assuming that all the parameters of structure-aware graph neural network $f(v, G)$ are independent and randomly drawn from Gaussian distributions, i.e., $\mathbf{W}^{(l)} \sim \mathcal{N}(\mathbf{0}, \sigma_w^2 \mathbf{I}), \mathbf{b}_{\text{SANN}}^{(l)} \sim \mathcal{N}(\mathbf{0}, \varsigma_b^2 \mathbf{I}), \mathbf{W}_{\text{SANN}}^{(l)} \sim \mathcal{N}(\mathbf{0}, \varsigma_w^2 \mathbf{I}), \tilde{\mathbf{b}}_{\text{SANN}}^{(l)} \sim \mathcal{N}(\mathbf{0}, \tilde{\varsigma}_b^2 \mathbf{I}), \tilde{\mathbf{W}}_{\text{SANN}}^{(l)} \sim \mathcal{N}(\mathbf{0}, \tilde{\varsigma}_w^2 \mathbf{I})$, when the layer width $M$ goes to infinity, the output function $f_i^{(l)}$ in Eq. (5) follows a Gaussian process with $f_i^{(l)} \sim \mathcal{GP}(0, K^{(l)})$, where the covariance function $K^{(l)}$ is given by*

$$K^{(l)} \left( (v, G), (v', G') \right) = \sigma_w^2 \cdot K_\mu^{(l)} \left( v, v' | G, G' \right) \cdot K_\nu^{(l)} \left( G, G' \right)$$

*where*

$$K_\mu^{(l)} \left( v, v' | G, G' \right) = \sum_{i,j=0}^{k} \alpha_i \alpha_j' \left( \varsigma_b^2 + \varsigma_w^2 \cdot \sum_{u \in N_i(v)} \sum_{u' \in N_j(v')} \mathbf{C}_{uu'}^{(l-1)}(\varsigma_w^2, \varsigma_b^2) \right)$$

$$K_\nu^{(l)} \left( G, G' \right) = \sum_{i,j=0}^{k} \alpha_i \alpha_j' \left( \tilde{\varsigma}_b^2 + \frac{\tilde{\varsigma}_w^2}{|V| \cdot |V'|} \mathbf{1}^T \mathbf{P}^{(i)} \mathbf{C}^{(l-1)}(\tilde{\varsigma}_w^2, \tilde{\varsigma}_b^2) \left( \mathbf{P}'^{(j)} \right)^T \mathbf{1} \right)$$

$$\mathbf{C}^{(l-1)}(a, b) = \mathbb{E}_{z_i^{(l-1)} \sim \mathcal{GP}\left(0, K_{ab}^{(l-1)}\right)} \left[ \phi(z_i^{(l-1)}) \phi(z_i^{(l-1)})^T \right]$$

*Here $\mathbf{P}^{(i)}$ ($\mathbf{P}'^{(j)}$) denotes the adjacency matrix given by the $i$-order neighborhood from graph $G$ ($j$-order neighborhood from graph $G'$). $K_{ab}^{(l-1)} = \sum_{i,j=0}^{k} \alpha_i \alpha_j' \left( b^2 + a^2 \mathbf{P}^{(i)} \mathbf{C}^{(l-2)}(a, b) (\mathbf{P}'^{(j)})^T \right)$ and $\mathbf{C}_{uu'}^{(0)}(a, b) = \langle \mathbf{x}_u, \mathbf{x}_{u'} \rangle$ for any $u \in V, u' \in V'$.*

### 4.2.1 Implications of Theorem 4.1

In the following, we show that the sample-level covariance/kernel $K_\mu^{(l)}$ of Theorem 4.1 can be explained as a message-passing operation in the kernel space. Following [4, 37], when using ReLU as activation function $\phi(\cdot)$ (i.e., $\phi(x) = \max\{0, x\}$), $\mathbf{C}^{(l-1)}$ is the arc-cosine kernel [12] as follows.

$$\mathbf{C}_{uu'}^{(l-1)}(\varsigma_w^2, \varsigma_b^2) = \frac{\kappa_1(\zeta)}{2}\sqrt{K_\mu^{(l-1)}(u, u|G) \cdot K_\mu^{(l-1)}(u', u'|G')}$$

where $\zeta = \frac{K_\mu^{(l-1)}(u, u'|G, G')}{\sqrt{K_\mu^{(l-1)}(u, u|G) \cdot K_\mu^{(l-1)}(u', u'|G')}}$ and $\kappa_1(\zeta) = \frac{1}{\pi}\left(\zeta \cdot (\pi - \arccos(\zeta)) + \sqrt{1 - \zeta^2}\right)$.

Then, the following corollary derives the feature map of $K_\mu^{(l)}(u, u'|G, G')$ in the kernel space.

**Corollary 4.2.** *Let $\varphi_1 : \mathcal{H} \to \mathcal{H}_1$ denote the kernel mapping from a pre-activation RKHS $\mathcal{H}$ to post-activation RKHS $\mathcal{H}_1$, i.e., $\langle \varphi_1(s), \varphi_1(s') \rangle = \frac{1}{2}||s|| \cdot ||s'|| \cdot \kappa_1\left(\frac{\langle s, s' \rangle}{||s|| \cdot ||s'||}\right)$. Given the sample-level kernel $K_\mu^{(l)}(v, v'|G, G')$ in Theorem 4.1, if the graph convolutional layer of Eq. (3) has no bias term ($\varsigma_b = 0$), the feature map of this kernel is given by*

$$\Psi_v^{(l)} = \varsigma_w \sum_{i=0}^k \alpha_i \sum_{u \in N_i(v)} \varphi_1\left(\Psi_u^{(l-1)}\right) \quad and \quad \Psi_{v'}^{(l)} = \varsigma_w \sum_{i=0}^k \alpha_i' \sum_{u' \in N_i(v')} \varphi_1\left(\Psi_{u'}^{(l-1)}\right)$$

*with $K_\mu^{(l)}(v, v'|G, G') = \langle \Psi_v^{(l)}, \Psi_{v'}^{(l)} \rangle$.*

Corollary 4.2 shows that the feature map $\Psi_v^{(l)}$ of sample-level kernel $K_\mu^{(l)}$ is a kernelized version of the graph convolutional layer in Eq. (3), which recursively aggregates message from nodes' neighborhood in the kernel space. The neighborhood can be automatically selected by optimizing the hyper-parameters $\alpha_i$ and $\alpha_i'$.

Next, we show the implication of domain-level kernel $K_\nu^{(l)}(G, G')$. As discussed in Subsection 4.1, a graph $G$, empirically drawn from graph domain distribution $\mathbb{P}_\mathcal{G}$, can approximate the true distribution $\mathbb{P}_\mathcal{G}$ when increasing the size of graph $G$. This motivates us to understand the connection between empirical domain representation $\nu^{(l)}(G)$ and the embedding of probability distribution $\mathbb{P}_\mathcal{G}$ [19, 48].

**Definition 4.3** (Mean Embedding of Probability Distributions [19]). Given an RKHS $\mathcal{H}$ induced by a kernel $k(\cdot, \cdot)$, the mean embedding of a probability distribution $\mathbb{P}$ is defined as $\tau_\mathbb{P} = \int k(\cdot, \mathbf{x}) d\mathbb{P}(\mathbf{x})$, i.e., $\mathbb{E}_{\mathbf{x} \sim \mathbb{P}}[f(\mathbf{x})] = \langle f, \tau_\mathbb{P} \rangle_\mathcal{H}$ for all $f \in \mathcal{H}$. Furthermore, the empirical mean embedding of $\mathbb{P}$ is given by $\hat{\tau}_\mathbb{P} = \frac{1}{m} \sum_{i=1}^m k(\cdot, \mathbf{x}_i)$, where $m$ samples are independent and identically drawn from $\mathbb{P}$.

Analogously, the mean embedding of a graph domain distribution $\mathbb{P}_\mathcal{G}$ can also be defined as $\tau_{\mathbb{P}_\mathcal{G}} = \int k_\mathcal{G}(\cdot, v|\mathbb{P}_\mathcal{G}) d\mathbb{P}_\mathcal{G}(v)$ given an RKHS $\mathcal{H}_\mathcal{G}$ induced by a kernel $k_\mathcal{G}(\cdot, \cdot)$. In the context of graph learning, $v|\mathbb{P}_\mathcal{G}$ indicates the individual node attributes of $v$ and graph structure information around node $v$ induced by $\mathbb{P}_\mathcal{G}$. It subsumes the conventional mean embedding [19] of a probability distribution $\tau_\mathbb{P}$ with independent and identically distributed (IID) samples $\mathbf{x} \sim \mathbb{P}$, if there are no edges in the graph (i.e., no graph structure is involved around node $v$). It is observed that the additional graph structure challenges empirical estimation of graph domain distribution $\mathbb{P}_\mathcal{G}$, thereby resulting in a nontrivial domain similarity estimator between source and target graphs in transferable graph learning. The following corollary states that the graph representation $\nu^{(l)}(G)$ can be considered as the empirical mean embedding of $\tau_{\mathbb{P}_\mathcal{G}}$ in the RKHS induced by $K_\nu^{(l)}(\cdot, \cdot)$.

**Corollary 4.4.** *With the same conditions in Theorem 4.1, for each $l$, in the limit on the layer width, the graph representation $\nu^{(l)}(G)$ recovers the empirical mean embedding $\hat{\tau}_{\mathbb{P}_\mathcal{G}}^{(l)}$ of the domain distribution $\mathbb{P}_\mathcal{G}$ in the reproducing kernel Hilbert space induced by $K_\nu^{(l)}$, and $\hat{\tau}_{\mathbb{P}_\mathcal{G}}^{(l)}$ is given by*

$$\hat{\tau}_{\mathbb{P}_\mathcal{G}} = \frac{1}{|V|} \sum_{v \in V} \tilde{\Psi}_v^{(l)} = \frac{1}{|V|} \sum_{v \in V} K_\nu^{(l)}(\cdot, v|G)$$

*where $\tilde{\Psi}_v^{(l)}$ is the feature map of $K_\nu^{(l)}$ with $\tilde{\Psi}_v^{(l)} = \tilde{\varsigma}_w \sum_{i=0}^k \alpha_i \sum_{u \in N_i(v)} \varphi_1\left(\tilde{\Psi}_u^{(l-1)}\right)$.*

Corollary 4.4 shows that $K_\nu^{(l)}(G, G')$ in Theorem 4.1 can be explained as the similarity of the empirical mean embeddings of domain distributions $\mathbb{P}_\mathcal{G}$ and $\mathbb{P}_{\mathcal{G}'}$ in the kernel space e.g., $K_\nu^{(l)}(G, G') = \langle \hat{\tau}_{\mathbb{P}_\mathcal{G}}^{(l)}, \hat{\tau}_{\mathbb{P}_{\mathcal{G}'}}^{(l)} \rangle_{\mathcal{H}_{K_\nu^{(l)}}}$. A similar metric over distributions is the maximum mean discrepancy (MMD) [19] defined over the distance of empirical mean embeddings, i.e., $\text{MMD}_\nu^{(l)} = ||\hat{\tau}_{\mathbb{P}_\mathcal{G}}^{(l)} - \hat{\tau}_{\mathbb{P}_{\mathcal{G}'}}^{(l)}||_{\mathcal{H}_{K_\nu^{(l)}}}$, where $\mathcal{H}_{K_\nu^{(l)}}$ denotes the RKHS induced by $K_\nu^{(l)}$. MMD has been widely applied to transfer learning [31, 32, 61] for measuring the distribution shift across image domains, under the covariate shift assumption [40] (i.e., source and target domains share conditional distribution $p(y|x)$ but different marginal distributions $p(x)$). In contrast, in this paper, we focus on measuring the domain distribution similarity for transferable graph learning.

### 4.2.2 Homophily vs. Heterophily

As illustrated in Section 3.2, there are two assumptions in graph learning [20, 68]: homophily and heterophily. We show the special cases of Theorem 4.1 in tackling homophily or heterophily based transferable graph learning. Given homophily source graph $G$ and target graph $G'$ where connected nodes have similar output values, it is revealed [20, 25, 59] that each node aggregates message from itself and its 1-order neighborhood, e.g., $k = 1$ and $\alpha_0 = \alpha_1 = \alpha_0' = \alpha_1' = 1$. Then the sample-level kernel $K_\mu^{(l)}(v, v'|G, G') \propto \sum_{u \in \{v \cup N_1(v)\}} \sum_{u' \in \{v' \cup N_1(v')\}} \mathbf{C}_{uu'}^{(l-1)}$. In contrast, if source and target graphs follow heterophily that connected nodes are not similar in the output space, nodes might aggregate message from high-order neighborhood [1, 52, 68], e.g., $k = 2$ and $\alpha_0 = \alpha_2 = 1, \alpha_1 = 0$. Then it holds that $K_\mu^{(l)}(v, v'|G, G') \propto \sum_{u \in \{v \cup N_2(v)\}} \sum_{u' \in \{v' \cup N_2(v')\}} \mathbf{C}_{uu'}^{(l-1)}$. These results indicate that by learning the neighborhood importances $\alpha_i$ and $\alpha_i'$, Gaussian process led by $f(v, G)$ can adaptively select the neighborhood for message aggregation from homophily or heterophily graphs. Moreover, $\alpha_i = \alpha_i'$ implies that source and target graphs follow the same assumption. The flexibility of $\alpha_i, \alpha_i'$ allows us to transfer knowledge across different types of graphs. For example, using $k = 2, \alpha_0 = \alpha_1 = 1, \alpha_2 = 0, \alpha_0' = \alpha_2' = 0, \alpha_1' = 0$, it enables the knowledge transferability from a homophily source graph to a heterophily target graph.

### 4.3 Proposed Algorithms

The goal of transferable node regression is to learn the prediction function for the target graph, by leveraging the input-output relationship from a relevant source graph. Given a source graph $G_s = (V_s, E_s)$ with fully labeled nodes (e.g., $y_v \in \mathbb{R}$ is associated with each node $v \in V_s$) and a target graph $G_t = (V_t, E_t)$ with a limited number of labeled nodes (e.g., $|V_t^{la}| \ll |V_s|$ where $V_t^{la} \subset V_t$ is the set of labeled target nodes), we propose the adaptive graph Gaussian process algorithm (termed as `GraphGP`) as follows.

For notation simplicity, we let $f_v = f^{(L)}(v, G)$ (given by $L$ graph convolutional layers in Eq. (5)) be the function value at node $v$. Let $\mathbf{f}_s = [f_1, f_2, \cdots, f_{|V_s|}]^T$ be a vector of latent function values over labeled source nodes, and $\mathbf{y}_s = [y_1, y_2, \cdot, y_{|V_s|}]^T$ be the associated ground-truth output values. Similarly, we can define $\mathbf{f}_t$ and $\mathbf{y}_t$ over target nodes. Then, the GP prior over function values can be defined as $f \sim \mathcal{GP}(0, K^{(L)}(\cdot, \cdot))$ (defined in Theorem 4.1), and its instantiation at labeled training nodes is given by $p(\mathbf{f}|V_s \cup V_t^{la}) = \mathcal{N}(\mathbf{0}, \mathbf{K}_{(s+t)(s+t)})$ where $\mathbf{K}_{(s+t)(s+t)}$ is a block matrix, i.e., $\mathbf{K}_{(s+t)(s+t)} = \begin{bmatrix} \mathbf{K}_{ss} & \mathbf{K}_{st} \\ \mathbf{K}_{ts} & \mathbf{K}_{tt} \end{bmatrix}$ and its entry is $[\mathbf{K}_{ab}]_{v_a v_b} = K^{(L)}((v_a, G_a), (v_b, G_b))$ for $a, b \in \{s, t\}$. For the likelihood, we consider the noisy scenarios where $p(\mathbf{y}_a|\mathbf{f}_a) = \mathcal{N}(\mathbf{f}_a, \varrho_a^2 \mathbf{I})$ for $a \in \{s, t\}$ ($\varrho_a$ measures the noisy magnitude). Then, following standard Gaussian process [42], the posterior distribution of `GraphGP` over testing nodes $V_* \subset V_t$ has a closed-form expression, i.e., $p(\mathbf{f}_*|V_*, \mathbf{f}, V_s \cup V_t^{la}) = \mathcal{N}(\boldsymbol{\gamma}, \boldsymbol{\Gamma})$ where

$$\boldsymbol{\gamma} = \mathbf{K}_* \left( \mathbf{K}_{(s+t)(s+t)} + \begin{bmatrix} \varrho_s^2 \mathbf{I} & \mathbf{0} \\ \mathbf{0} & \varrho_t^2 \mathbf{I} \end{bmatrix} \right)^{-1} \mathbf{y} \quad \boldsymbol{\Gamma} = \mathbf{K}_{**} - \mathbf{K}_* \left( \mathbf{K}_{(s+t)(s+t)} + \begin{bmatrix} \varrho_s^2 \mathbf{I} & \mathbf{0} \\ \mathbf{0} & \varrho_t^2 \mathbf{I} \end{bmatrix} \right)^{-1} \mathbf{K}_*^T \quad (6)$$

Here, $\mathbf{y} = \begin{bmatrix} \mathbf{y}_s \\ \mathbf{y}_{t_{la}} \end{bmatrix}$ denotes the ground-truth output values of labeled nodes $V_s \cup V_t^{la}$ from source and target graphs. Each entry of $\mathbf{K}_*$ represents the covariance between testing target node and training node, and $\mathbf{K}_{**}$ denotes the covariance matrix over testing target nodes.

The posterior distribution of `GraphGP` has the following hyper-parameters: $\sigma_w, \varsigma_w, \varsigma_b, \alpha_i^s, \alpha_j^t$, and noise variances $\varrho_s, \varrho_t$. The hyper-parameters of the Gaussian process can be optimized by maximizing the marginal likelihood $p(\mathbf{y})$ over all the training samples [42]. However, in the context of transferable graph learning, the number of labeled target nodes is much smaller than the number of labeled source nodes. By directly maximizing the marginal likelihood over all labeled source and target nodes, `GraphGP` might be biased towards the source domain. Therefore, in this paper, we propose to optimize the marginal distribution $p(\mathbf{y}_{t_{la}})$ over labeled target nodes by considering all the nodes $V_s \cup V_t^{la}$ as the inducing points [47] (see more efficiency analysis in Appendix A.6). The objective function is given by maximizing the following log marginal likelihood

$$\log p(\mathbf{y}_{t_{la}}) = \log \left[ \mathcal{N} \left( \mathbf{y}_{t_{la}} | \mathbf{0}, \mathbf{K}_{t(s+t)} \mathbf{K}_{(s+t)(s+t)}^{-1} \mathbf{K}_{t(s+t)}^T + \varrho_t^2 \mathbf{I} \right) \right] \tag{7}$$

### 4.4 Generalization Analysis

We define the generalization error of the target graph for transferable node regression. Given the ground-truth labeling function $f^*$ in the target domain and the estimator $f^{(L)}$ learned from observed source and target graphs, the generalization error is given by

$$\epsilon_t = \int \int \mathcal{L} \left( \gamma(v_t, G_t), f^*(v_t, G_t) \right) p(v_t, G_t) p(f^*) d(v_t, G_t) df^* \tag{8}$$

where $\gamma(\cdot)$ is the posterior mean of $f^{(L)}(\cdot)$ in Eq. (6), $\mathcal{L}$ is the mean squared loss function, and $p(v_t, G_t)$ denotes the sampling probability[2] of a target input $(v_t, G_t)$. It is shown [9, 42] that if the GP prior is correctly specified, i.e., the predictor $f^{(L)}$ has the same prior as $f^*$, the generalization error is given by the predictive variance of the GP. That is, $\epsilon_t = \int \Gamma(v_t, G_t) p(v_t, G_t) d(v_t, G_t)$ where $\Gamma(v_t, G_t)$ is the posterior variance at $(v_t, G_t)$ in Eq. (6). Furthermore, the following theorem shows that the generalization error can be upper bounded in terms of the variance that takes all inputs $(v_s, G_s)$ of the source graph as the target samples, i.e., both $(v_s, G_s)$ and $(v_t, G_t)$ are assumed to follow the same target distribution for single-domain graph learning.

**Theorem 4.5.** *Let $\mathbf{K}_{ss}^\mu$ be the sample-level covariance matrix over source nodes, i.e., $[\mathbf{K}_{ss}^\mu]_{(v_s, v_s')} = K_\mu^{(L)}(v_s, v_s'|G_s)$, $\nu_{ss} = K_\nu^{(L)}(G_s, G_s), \nu_{tt} = K_\nu^{(L)}(G_t, G_t)$ be the intra-graph kernels and $\nu_{st} = K_\nu^{(L)}(G_s, G_t)$ be the inter-graph kernel. Suppose $\bar{\varrho}_s^2 \triangleq \left( \frac{\nu_{ss} \cdot \nu_{tt}}{\nu_{st} \cdot \nu_{st}} - 1 \right) \sigma_w^2 \nu_{tt} \bar{\lambda}_{ss} + \frac{\nu_{tt} \cdot \nu_{tt}}{\nu_{st} \cdot \nu_{st}} \varrho_s^2$ where $\bar{\lambda}_{ss}$ is the maximum eigenvalue of $\mathbf{K}_{ss}^\mu$, for any $(v_t, G_t)$ the generalization error is bounded by*

$$\epsilon_t \leq \int \Gamma_t(v_t, G_t; \bar{\varrho}_s^2, \varrho_t^2) p(v_t, G_t) d(v_t, G_t)$$

*where $\Gamma_t(v_t, G_t; \bar{\varrho}_s^2, \varrho_t^2)$ is the variance assuming that all source examples are observed in the target domain with respect to noises $\bar{\varrho}_s^2, \varrho_t^2$.*

This theorem confirms that compared to the single-domain Gaussian process (assuming all inputs are drawn from the same distribution), `GraphGP`-based transferable graph learning enables the reduced generalization error. In addition, the following corollary considers a special scenario where no labeled nodes are available in the target graph. In this case, the generalization error is determined by the normalized graph domain similarity $\frac{K_\nu^{(L)}(G_s, G_t)}{\sqrt{K_\nu^{(L)}(G_s, G_s) \cdot K_\nu^{(L)}(G_t, G_t)}}$.

**Corollary 4.6.** *When there are no labeled nodes in the target graph, i.e., $V_t^{la} = \emptyset$, we have*

$$\lim_{|V_s| \to \infty} \epsilon_t = \left( 1 - \frac{\nu_{st}^2}{\nu_{tt} \nu_{ss}} \right) \int K^{(L)} \left( (v_t, G_t), (v_t, G_t) \right) p(v_t, G_t) d(v_t, G_t)$$

*where $\nu_{ss} = K_\nu^{(L)}(G_s, G_s), \nu_{tt} = K_\nu^{(L)}(G_t, G_t)$ and $\nu_{st} = K_\nu^{(L)}(G_s, G_t)$.*

---

[2] Here, it holds $p(v_t, G_t) = \mathbb{P}_{\mathscr{P}}(\mathbb{P}_{\mathcal{G}_t}) \mathbb{P}_{\mathcal{G}_t}(v_t)$ as discussed in Subsection 4.1, i.e., it first samples a graph domain distribution $\mathbb{P}_{\mathcal{G}_t}$ and then sample the nodes/edges to form the graph $G_t$.

| Model | Airport graphs | | | Agriculture graphs | | |
|---|---|---|---|---|---|---|
| | BR → EU | EU → BR | BR → US | MA → SG | SG → MA | MA → SY |
| RBFGP [42] | $0.4849_{\pm 0.0260}$ | $0.4479_{\pm 0.0211}$ | $0.3682_{\pm 0.0301}$ | $0.4859_{\pm 0.0716}$ | $0.3359_{\pm 0.0297}$ | $0.7314_{\pm 0.0172}$ |
| DINO [56] | $0.5241_{\pm 0.0147}$ | $0.4855_{\pm 0.0337}$ | $0.3877_{\pm 0.0263}$ | $0.6227_{\pm 0.0223}$ | $0.4591_{\pm 0.0445}$ | $0.7620_{\pm 0.0125}$ |
| GGP [35] | $0.3400_{\pm 0.0144}$ | $0.3990_{\pm 0.0401}$ | $0.4720_{\pm 0.0218}$ | $0.4515_{\pm 0.0099}$ | $0.2403_{\pm 0.0172}$ | $0.7420_{\pm 0.0425}$ |
| SAGEGP [37] | $0.4581_{\pm 0.0308}$ | $0.3822_{\pm 0.0508}$ | $0.4928_{\pm 0.0330}$ | $0.5348_{\pm 0.0278}$ | $0.3633_{\pm 0.0101}$ | $0.7632_{\pm 0.0349}$ |
| GINGP [37] | $0.5216_{\pm 0.0227}$ | $0.4471_{\pm 0.0219}$ | $0.4901_{\pm 0.0255}$ | $0.5380_{\pm 0.0288}$ | $0.3559_{\pm 0.0196}$ | $0.7746_{\pm 0.0518}$ |
| GraphGP | $\mathbf{0.5567_{\pm 0.0246}}$ | $\mathbf{0.4983_{\pm 0.0370}}$ | $\mathbf{0.5293_{\pm 0.0335}}$ | $\mathbf{0.6586_{\pm 0.0244}}$ | $\mathbf{0.5125_{\pm 0.0171}}$ | $\mathbf{0.7921_{\pm 0.0168}}$ |

Table 1: Results of transferable node regression on Airport and Agriculture data sets

| Model | PT → RU | EN → PT | RU → ES | RU → PT | PT → EN | ES → RU |
|---|---|---|---|---|---|---|
| RBFGP [42] | $0.4324_{\pm 0.0087}$ | $0.5430_{\pm 0.0117}$ | $0.4241_{\pm 0.0024}$ | $0.4962_{\pm 0.0055}$ | $0.5507_{\pm 0.0024}$ | $0.4738_{\pm 0.0059}$ |
| DINO [56] | $0.6498_{\pm 0.0059}$ | $0.7565_{\pm 0.0140}$ | $0.6717_{\pm 0.0114}$ | $0.7641_{\pm 0.0082}$ | $0.5920_{\pm 0.0195}$ | $0.5636_{\pm 0.0072}$ |
| GGP [35] | $0.3167_{\pm 0.0079}$ | $0.5689_{\pm 0.0028}$ | $0.3699_{\pm 0.0087}$ | $0.5173_{\pm 0.0018}$ | $0.4237_{\pm 0.0037}$ | $0.4237_{\pm 0.0035}$ |
| SAGEGP [37] | $0.5655_{\pm 0.0196}$ | $0.5191_{\pm 0.0029}$ | $0.5958_{\pm 0.0076}$ | $0.6921_{\pm 0.0174}$ | $0.5720_{\pm 0.0187}$ | $0.5960_{\pm 0.0074}$ |
| GINGP [37] | $0.5782_{\pm 0.0218}$ | $0.7525_{\pm 0.0294}$ | $0.6803_{\pm 0.0198}$ | $0.6910_{\pm 0.0540}$ | $0.6349_{\pm 0.0090}$ | $0.5997_{\pm 0.0185}$ |
| GraphGP | $\mathbf{0.7069_{\pm 0.0055}}$ | $\mathbf{0.8013_{\pm 0.0109}}$ | $\mathbf{0.7520_{\pm 0.0144}}$ | $\mathbf{0.7909_{\pm 0.0382}}$ | $\mathbf{0.6745_{\pm 0.0127}}$ | $\mathbf{0.6789_{\pm 0.0172}}$ |

Table 2: Results of transferable node regression on Twitch data set

| Model | Wikipedia graphs | | WebKB graphs | |
|---|---|---|---|---|
| | SQ → CH | CH → SQ | CO → TX | WS → TX |
| RBFGP | $-0.0383_{\pm 0.0526}$ | $-0.0119_{\pm 0.0288}$ | $0.3089_{\pm 0.0533}$ | $0.2756_{\pm 0.0204}$ |
| DINO | $0.2938_{\pm 0.0253}$ | $0.1157_{\pm 0.0145}$ | $0.3536_{\pm 0.0442}$ | $0.2537_{\pm 0.0232}$ |
| LINKXGP | $0.4100_{\pm 0.0205}$ | $0.0838_{\pm 0.0307}$ | $0.3836_{\pm 0.0530}$ | $0.2898_{\pm 0.0176}$ |
| MixHopGP | $0.4050_{\pm 0.0456}$ | $\mathbf{0.3509_{\pm 0.0102}}$ | $0.3264_{\pm 0.0569}$ | $0.3062_{\pm 0.0337}$ |
| H2GCNGP | $0.4165_{\pm 0.0559}$ | $0.2652_{\pm 0.0106}$ | $0.3816_{\pm 0.0430}$ | $0.3036_{\pm 0.0442}$ |
| GraphGP | $\mathbf{0.4938_{\pm 0.0352}}$ | $0.3214_{\pm 0.0080}$ | $\mathbf{0.4146_{\pm 0.0402}}$ | $\mathbf{0.3301_{\pm 0.0585}}$ |

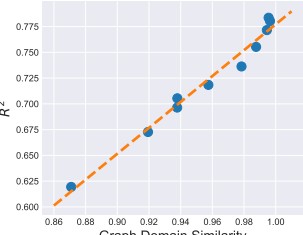

Table 3: Results on Wikipedia and WebKB        Figure 2: Domain similarity

## 5 Experiments

**Data Sets:** We use the following graph learning benchmarks with node regression tasks. (1) Twitch [44]: It has 6 different domains ("DE", "EN", "ES", "FR", "PT", and "RU"). (2) Agriculture [34, 60]: It has 3 different domains ("Maize" (MA), "Sorghum" (SG), and "Soybean" (SY)). (3) Airports [43]: It has 3 different domains ("USA" (US), "Brazil" (BR), and "Europe" (EU)). (4) Wikipedia [44]: It has 3 different domains ("chameleon" (CH), "crocodile" (CR), and "squirrel" (SQ)). (5) WebKB [41]: It has 3 different domains ("Cornell" (CO), "Texas" (TX), and "Wisconsin" (WS)).

**Baselines:** We consider the following Gaussian process baselines. (1) RBFGP [42] and DINO [56] are feature-Only Gaussian processes without using graph structures. (2) GGP [35], SAGEGP [37], and GINGP [37] are graph Gaussian processes by considering source and target graphs as a large disjoint graph. (3) LINKXGP, MixHopGP, and H2GCNGP are graph Gaussian processes derived from LINKX [30], MixHop [1], H2GCN [68] respectively. It is notable that recent work [37] shows the equivalence between graph neural networks (e.g., GraphSAGE [20], GIN [59]) and graph Gaussian processes. Similarly, we can derive the corresponding graph Gaussian processes for LINKX [30], MixHop [1] and H2GCN [68] in Appendix A.5, which are termed as LINKXGP, MixHopGP, and H2GCNGP, respectively.

**Model Configuration:** In the experiments, we use GPyTorch [16] to build the graph Gaussian process models and optimize the hyperparameters with gradient descent optimizer. For the proposed GraphGP algorithm, we adopt $k = 2$ and $L = 2$ for all the experiments. The hyperparameters are optimized using Adam [24] with a learning rate of 0.01 and a total number of training epochs of 500. All the experiments are performed on a Windows machine with four 3.80GHz Intel Cores, 64GB RAM, and two NVIDIA Quadro RTX 5000 GPUs[3].

### 5.1 Results

---

[3]Code is available at https://github.com/jwu4sml/GraphGP.

Table 1 and Table 2 provide the transferable node regression results on Airport, Agriculture, and Twitch graphs. Following [37], we report the coefficient of determination $R^2$ (mean and standard deviation with 5 runs) on the testing target nodes for performance comparison. The experimental results verify the effectiveness of GraphGP over Gaussian process baselines. It is notable that the graphs in Table 1 and Table 2 follow the homophily assumption. In contrast, Wikipedia and WebKB graphs in Table 3 hold the heterophily assumption [68]. It can be seen that under heterophily assumption,

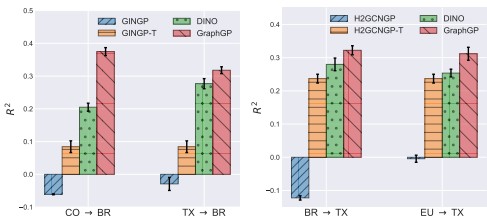

(a) WebKB → Airport  (b) Airport → WebKB

Figure 3: Knowledge transfer across graphs with different assumptions

GraphGP also archives much better performance in most cases. In addition, we evaluate the knowledge transferability across graphs with different assumptions in Figure 3. It is observed that compared to GINGP (or H2GCNGP) over both labeled source and target nodes, GINGP-T (or H2GCNGP-T) learns the Gaussian process over only target nodes, thereby enabling much better transfer performance from WebKB to Airport in Figure 3a (or from Airport to WebKB in Figure 3b). Moreover, GraphGP can outperform those target-only graph Gaussian processes in this scenario.

We also investigate the correlation between transfer performance and normalized graph domain similarity $\frac{K_\nu^{(L)}(G_s, G_t)}{\sqrt{K_\nu^{(L)}(G_s, G_s) \cdot K_\nu^{(L)}(G_t, G_t)}}$ in Corollary 4.6. Figure 2 visualizes the estimated normalized graph domain similarity of GraphGP and $R^2$ over the target testing nodes for RU → PT on Twitch, where the graph domain similarity changes when injecting noise to the target graph (see Appendix A.7.1). It shows that the transfer performance is positively correlated with the normalized graph domain similarity. This is consistent with our observation in Corollary 4.6.

## 5.2 Analysis

**Flexibility of GraphGP:** As illustrated in Subsection 4.1, it is flexible to instantiate structure-aware neural networks of Eq. (2) with existing message-passing graph neural networks. Table 4 provides the results of GIN [59] induced GraphGP algorithm. Furthermore, it is feasible to simply instantiate $K_\nu^{(L)}(G, G')$ of GraphGP with existing graph kernels, e.g., Shortest Path kernel [5]. It is observed that the variants GraphGP_GIN and GraphGP_ShortestPath of

| Methods | BR → EU |
|---|---|
| GraphGP | $0.5567_{\pm 0.0246}$ |
| GraphGP_GIN | $0.5301_{\pm 0.0229}$ |
| GraphGP_ShortestPath | $0.5377_{\pm 0.0304}$ |

Table 4: Flexibility of GraphGP

GraphGP achieve comparable performance. This highlights that GraphGP is flexible to incorporate with existing GNNs and graph kernels for transferable graph learning tasks.

**Comparison with Transferable GNNs:** In addition to Gaussian processes, we also compare GraphGP with state-of-the-art transferable graph neural networks. Table 5 shows the competitive performance of GraphGP over transferable GNN baselines. This is because GraphGP explicitly maximizes the marginal likelihood $p(\mathbf{y}_{t_{la}})$.

| Methods | BR → EU | EU → BR | BR → US |
|---|---|---|---|
| EGI [69] | $0.5204_{\pm 0.0357}$ | $0.4786_{\pm 0.0225}$ | $0.4951_{\pm 0.0176}$ |
| GARDE [55] | $0.5314_{\pm 0.0208}$ | $0.4792_{\pm 0.0296}$ | $0.4354_{\pm 0.0109}$ |
| GraphGP | $0.5567_{\pm 0.0246}$ | $0.4983_{\pm 0.0370}$ | $0.5293_{\pm 0.0335}$ |

Table 5: Performance comparison between GraphGP and transferable GNNs

In contrast, transferable GNN baselines minimize the prediction loss over all the labeled nodes ($|V_s| \gg |V_t^{la}|$) and domain discrepancy, and thus they are more likely to bias towards the source graph.

## 6 Conclusion

This paper studies the problem of transferable graph learning involving knowledge transfer from a source graph to a relevant target graph. To solve this problem, we propose a graph Gaussian process (GraphGP) algorithm, which is derived from a structure-aware neural network encoding both sample-level node representation and domain-level graph representation. The efficacy of GraphGP is verified theoretically and empirically in various transferable node regression tasks.

## Acknowledgments and Disclosure of Funding

This work is supported by National Science Foundation under Award No. IS-1947203, IIS-2117902, IIS-2137468, IIS-2002540, and Agriculture and Food Research Initiative (AFRI) grant no. 2020-67021-32799/project accession no.1024178 from the USDA National Institute of Food and Agriculture. The views and conclusions are those of the authors and should not be interpreted as representing the official policies of the funding agencies or the government.

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

# A Appendix

In the appendix, we provide more details regarding the proposed `GraphGP` for transferable graph learning in the paper, including

- In Subsections A.1 and A.2, we discuss the broader impacts and limitations of this paper.
- In Subsection A.3, we prove the theorems and corollaries stated in the paper.
- In Subsection A.4, we discuss the instantiations of our `GraphGP` algorithm with existing graph neural networks.
- In Subsection A.5, we derive the graph Gaussian Processes of existing Heterophilic GNNs, including LINKX [30], MixHop [1], H2GCN [68]. These derived Gaussian Processes are used as the baselines in our experiments (see Table 3).
- In Subsection A.6, we discuss the model efficiency of `GraphGP` and provide a computationally efficient approximation solution for `GraphGP`based on Nyström approximation.
- In Subsection A.7, we provide detailed experimental setups and additional evaluation results.

## A.1 Broader Impacts

This paper focuses on the fundamental research problem of transferable graph learning. The goal is to effectively understand a target graph by leveraging latent knowledge from a relevant source graph. Generally, there are no negative societal impacts involved in this work.

## A.2 Limitations

This paper develops a generic framework to build graph Gaussian processes for transferable graph learning under covariate shift assumption [40] over graphs. In addition to covariate shift, label shift (i.e., $\mathbb{P}_{\mathcal{Y}}$ shifts across domains) is also commonly considered in transfer learning scenarios. It is much more challenging to extend the developed transferable graph Gaussian processes to tackle label shift scenarios. This is because it is difficult to accurately estimate the graph domain distribution with limited label information from the target domain. In addition, following standard transfer learning [40], after model training, we simply apply the learned posterior function over testing target nodes for predicting their output values. It might be feasible to incorporate `GraphGP` with test-time adaptation techniques [51] to further improve the transfer performance.

## A.3 Proof of Theorems and Corollaries

### A.3.1 Proof of Theorem 4.1

Theorem 4.1 states that assume all the parameters of structure-aware graph neural network $f(v, G)$ are independent and randomly drawn from Gaussian distributions, i.e., $\mathbf{W}^{(l)} \sim \mathcal{N}(\mathbf{0}, \sigma_w^2 \mathbf{I})$, $\mathbf{b}_{\text{SANN}}^{(l)} \sim \mathcal{N}(\mathbf{0}, \varsigma_b^2 \mathbf{I})$, $\mathbf{W}_{\text{SANN}}^{(l)} \sim \mathcal{N}(\mathbf{0}, \varsigma_w^2 \mathbf{I})$, $\tilde{\mathbf{b}}_{\text{SANN}}^{(l)} \sim \mathcal{N}(\mathbf{0}, \tilde{\varsigma}_b^2 \mathbf{I})$, $\tilde{\mathbf{W}}_{\text{SANN}}^{(l)} \sim \mathcal{N}(\mathbf{0}, \tilde{\varsigma}_w^2 \mathbf{I})$, when the layer width $M$ goes to infinity, the output function $f_i^{(l)}$ in Eq. (5) follows a Gaussian process with $f_i^{(l)} \sim \mathcal{GP}(0, K^{(l)})$, where the covariance function $K^{(l)}$ is given by

$$K^{(l)}\left((v, G), (v', G')\right) = \sigma_w^2 \cdot K_\mu^{(l)}\left(v, v' | G, G'\right) \cdot K_\nu^{(l)}\left(G, G'\right)$$

where

$$K_\mu^{(l)}\left(v, v' | G, G'\right) = \sum_{i,j=0}^{k} \alpha_i \alpha_j' \left( \varsigma_b^2 + \varsigma_w^2 \cdot \sum_{u \in N_i(v)} \sum_{u' \in N_j(v')} \mathbf{C}_{uu'}^{(l-1)}(\varsigma_w^2, \varsigma_b^2) \right)$$

$$K_\nu^{(l)}\left(G, G'\right) = \sum_{i,j=0}^{k} \alpha_i \alpha_j' \left( \tilde{\varsigma}_b^2 + \frac{\tilde{\varsigma}_w^2}{|V| \cdot |V'|} \mathbf{1}^T \mathbf{P}^{(i)} \mathbf{C}^{(l-1)}(\tilde{\varsigma}_w^2, \tilde{\varsigma}_b^2) \left(\mathbf{P}'^{(j)}\right)^T \mathbf{1} \right)$$

$$\mathbf{C}^{(l-1)}(a, b) = \mathbb{E}_{z_i^{(l-1)} \sim \mathcal{GP}\left(0, K_{ab}^{(l-1)}\right)} \left[ \phi(z_i^{(l-1)}) \phi(z_i^{(l-1)})^T \right]$$

Here $\mathbf{P}^{(i)}$ ($\mathbf{P}'^{(j)}$) denotes the adjacent matrix given by the $i$-order neighborhood from graph $G$ ($j$-order neighborhood from graph $G'$). $K_{ab}^{(l-1)} = \sum_{i,j=0}^{k} \alpha_i \alpha_j' \left( b^2 + a^2 \mathbf{P}^{(i)} \mathbf{C}^{(l-2)}(a,b)(\mathbf{P}'^{(j)})^T \right)$ and $\mathbf{C}_{uu'}^{(0)}(a,b) = \langle \mathbf{x}_u, \mathbf{x}_{u'} \rangle$ for any $u \in V, u' \in V'$.

*Proof.* Given the definition of $f(v,G)$, we have

$$\mathbf{h}_v^{(l)} = \sum_{i=0}^{k} \alpha_i \left( \frac{1}{\sqrt{M}} \sum_{u \in N_i(v)} \mathbf{W}_{\text{SANN}}^{(l-1)} \mathbf{x}_u^{(l)} + \mathbf{b}_{\text{SANN}}^{(l-1)} \right) \quad \text{and} \quad \mathbf{x}_u^{(l)} = \phi\left( \mathbf{h}_u^{(l-1)} \right)$$

$$\tilde{\mathbf{h}}_v^{(l)} = \sum_{i=0}^{k} \alpha_i \left( \frac{1}{\sqrt{M}} \sum_{u \in N_i(v)} \tilde{\mathbf{W}}_{\text{SANN}}^{(l-1)} \tilde{\mathbf{x}}_u^{(l)} + \tilde{\mathbf{b}}_{\text{SANN}}^{(l-1)} \right) \quad \text{and} \quad \tilde{\mathbf{x}}_u^{(l)} = \phi\left( \tilde{\mathbf{h}}_u^{(l-1)} \right)$$

where $\mathbf{x}_u^{(0)} = \tilde{\mathbf{x}}_u^{(0)} = \mathbf{x}_u$ denote the initial node attributes. The $l$-th layer of the function $f(v,G)$ is defined as

$$f_i^{(l)}(v,G) = \frac{1}{\sqrt{M}} \sum_{j=1}^{M} \mathbf{W}_{ij}^{(l)} \cdot \mu_j^{(l)}(v|G) \cdot \nu_j^{(l)}(G)$$

$$\text{where} \quad \mu^{(l)}(v|G) = \mathbf{h}_v^{(l)} \quad \text{and} \quad \nu^{(l)}(G) = \frac{1}{|V|} \sum_{v \in V} \tilde{\mathbf{h}}_v^{(l)}$$

Thus, we obtain $\mathbb{E}[f_i^{(l)}(v,G)] = 0$ and

$$K^{(l)}\left((v,G),(v',G')\right) \equiv \mathbb{E}[f_i^{(l)}(v,G) f_i^{(l)}(v',G')]$$

$$= \sigma_w^2 \cdot \mathbb{E}\left[ \mu_i^{(l)}(v|G) \cdot \mu_i^{(l)}(v'|G') \right] \cdot \mathbb{E}\left[ \nu_i^{(l)}(G) \cdot \nu_i^{(l)}(G') \right]$$

Moreover,

$$K_\mu^{(l)}\left(v,v'|G,G'\right) \equiv \mathbb{E}\left[ \mu_i^{(l)}(v|G) \cdot \mu_i^{(l)}(v'|G') \right]$$

$$= \mathbb{E}\left[ [\mathbf{h}_v^{(l)}]_i \cdot [\mathbf{h}_{v'}^{(l)}]_i \right]$$

$$= \sum_{i,j=0}^{k} \alpha_i \alpha_j' \left( \varsigma_b^2 + \varsigma_w^2 \cdot \mathbb{E}\left[ \sum_{u \in N_i(v)} [\mathbf{x}_u^{(l)}]_i \sum_{u' \in N_j(v')} [\mathbf{x}_{u'}^{(l)}]_i \right] \right)$$

$$= \sum_{i,j=0}^{k} \alpha_i \alpha_j' \left( \varsigma_b^2 + \varsigma_w^2 \cdot \sum_{u \in N_i(v)} \sum_{u' \in N_j(v')} \mathbb{E}\left[ [\mathbf{x}_u^{(l)}]_i [\mathbf{x}_{u'}^{(l)}]_i \right] \right)$$

$$= \sum_{i,j=0}^{k} \alpha_i \alpha_j' \left( \varsigma_b^2 + \varsigma_w^2 \cdot \sum_{u \in N_i(v)} \sum_{u' \in N_j(v')} \mathbf{C}_{uu'}^{(l-1)}(\varsigma_w^2, \varsigma_b^2) \right)$$

where $[\mathbf{a}]_i$ denotes the $i$-th entry of vector $\mathbf{a}$. Besides,

$$K_\nu^{(l)}\left(G,G'\right) \equiv \mathbb{E}\left[ \nu_i^{(l)}(G) \cdot \nu_i^{(l)}(G') \right]$$

$$= \mathbb{E}\left[ \frac{1}{|V|} \sum_{v \in V} [\tilde{\mathbf{h}}_v^{(l)}]_i \cdot \frac{1}{|V'|} \sum_{v' \in V'} [\tilde{\mathbf{h}}_{v'}^{(l)}]_i \right]$$

$$= \frac{1}{|V| \cdot |V'|} \sum_{v \in V} \sum_{v' \in V'} \mathbb{E}\left[ [\tilde{\mathbf{h}}_v^{(l)}]_i \cdot [\tilde{\mathbf{h}}_{v'}^{(l)}]_i \right]$$

$$= \frac{1}{|V| \cdot |V'|} \sum_{v \in V} \sum_{v' \in V'} \left( \sum_{i,j=0}^{k} \alpha_i \alpha_j' \left( \tilde{\varsigma}_b^2 + \tilde{\varsigma}_w^2 \cdot \sum_{u \in N_i(v)} \sum_{u' \in N_j(v')} \mathbf{C}_{uu'}^{(l-1)}(\tilde{\varsigma}_w^2, \tilde{\varsigma}_b^2) \right) \right)$$

$$= \sum_{i,j=0}^{k} \alpha_i \alpha_j' \left( \tilde{\varsigma}_b^2 + \frac{\tilde{\varsigma}_w^2}{|V| \cdot |V'|} \mathbf{1}^T \mathbf{P}^{(i)} \mathbf{C}^{(l-1)}(\tilde{\varsigma}_w^2, \tilde{\varsigma}_b^2) \left( \mathbf{P}'^{(j)} \right)^T \mathbf{1} \right)$$

which completes the proof. $\qquad\square$

### A.3.2 Proof of Positive Definiteness

With the assumptions in Theorem 4.1, the covariance kernel $K^{(l)}$ is positive definite.

*Proof.* This corollary can be proven using the Schur Product theorem that the Hadamard product of two positive semidefinite matrices is also a positive semidefinite matrix. Based on the definition of $(v, v'|G, G')$ and $K_\nu^{(l)}(G, G')$, the covariance matrix of these kernels over all nodes in $G, G'$ are positive semidefinite. Then as explained in [37], a kernel function is positive definite (resp. strictly positive definite) if the corresponding kernel matrix is positive semi-definite (resp. positive definite) for any collection of distinct points. Therefore, the kernel function $K^{(l)}((v, G), (v', G'))$ is symmetric and positive definite. Then based on Moore–Aronszajn theorem [2], $K^{(l)}$ defines a unique reproducing kernel Hilbert space (RKHS). $\qquad\square$

### A.3.3 Proof of Corollary 4.2

Corollary 4.2 states let $\varphi_1 : \mathcal{H} \to \mathcal{H}_1$ denote the kernel mapping from a pre-activation RKHS $\mathcal{H}$ to post-activation RKHS $\mathcal{H}_1$, i.e., $\langle \varphi_1(s), \varphi_1(s') \rangle = \frac{1}{2} ||s|| \cdot ||s'|| \cdot \kappa_1 \left( \frac{\langle s, s' \rangle}{||s|| \cdot ||s'||} \right)$. Given the sample-level kernel $K_\mu^{(l)}(v, v'|G, G')$ in Theorem 4.1, if the graph convolutional layer of Eq. (3) has no bias term ($\varsigma_b = 0$), the feature map of this kernel is given by[4]

$$\Psi_v^{(l)} = \varsigma_w \sum_{i=0}^{k} \alpha_i \sum_{u \in N_i(v)} \varphi_1 \left( \Psi_u^{(l-1)} \right) \quad \text{and} \quad \Psi_{v'}^{(l)} = \varsigma_w \sum_{i=0}^{k} \alpha_i' \sum_{u' \in N_i(v')} \varphi_1 \left( \Psi_{u'}^{(l-1)} \right)$$

with $K_\mu^{(l)}(v, v'|G, G') = \langle \Psi_v^{(l)}, \Psi_{v'}^{(l)} \rangle$.

*Proof.* Based on Theorem A.1, we have

$$\mathbf{C}_{vv'}^{(l-1)} = \frac{\kappa_1(\zeta)}{2} \sqrt{K_\mu^{(l-1)}(v, v|G) \cdot K_\mu^{(l-1)}(v', v'|G')}$$

$$= \left\langle \varphi_1 \left( \Psi_v^{(l-1)} \right), \varphi_1 \left( \Psi_{v'}^{(l-1)} \right) \right\rangle$$

Then, with $\varsigma_b = 0$, we have

$$K_\mu^{(l)}(v, v'|G, G') = \sum_{i,j=0}^{k} \alpha_i \alpha_j' \left( \varsigma_w^2 \cdot \sum_{u \in N_i(v)} \sum_{u' \in N_j(v')} \mathbf{C}_{uu'}^{(l-1)}(\varsigma_w^2, \varsigma_b^2) \right)$$

$$= \sum_{i,j=0}^{k} \alpha_i \alpha_j' \left( \varsigma_w^2 \cdot \sum_{u \in N_i(v)} \sum_{u' \in N_j(v')} \left\langle \varphi_1 \left( \Psi_u^{(l-1)} \right), \varphi_1 \left( \Psi_{u'}^{(l-1)} \right) \right\rangle \right)$$

$$= \left\langle \varsigma_w \sum_{i=0}^{k} \alpha_i \sum_{u \in N_i(v)} \varphi_1 \left( \Psi_u^{(l-1)} \right), \ \varsigma_w \sum_{j=0}^{k} \alpha_j' \sum_{u' \in N_j(v')} \varphi_1 \left( \Psi_{u'}^{(l-1)} \right) \right\rangle$$

Therefore, it holds that

$$\Psi_v^{(l)} = \varsigma_w \sum_{i=0}^{k} \alpha_i \sum_{u \in N_i(v)} \varphi_1 \left( \Psi_u^{(l-1)} \right)$$

$$\Psi_{v'}^{(l)} = \varsigma_w \sum_{j=0}^{k} \alpha_j' \sum_{u' \in N_j(v')} \varphi_1 \left( \Psi_{u'}^{(l-1)} \right)$$

This result provides the feature map for nodes from graph $G$ and $G'$ separately. But we show that it is equivalent to the following unified feature mapping on a large graph integrating $G$ and $G'$.

---

[4]For simplicity, we show the results without bias term, but it can be easily generalized to Eq. (3) with bias by absorbing the bias term in the kernel mapping.

More specifically, graphs $G = (V, E)$ and $G' = (V', E')$ can be considered as a single large graph $G'' = (V \cup V', E \cup E')$ with two disjoint components. Then, for any node $v'' \in V''$ in graph $G''$, we have

$$\Psi_{v''}^{(l)} = \varsigma_w \sum_{i=0}^{k} (\alpha_i \cdot \mathbb{I}[v'' \in V] + \alpha_i' \cdot \mathbb{I}[v'' \in V']) \sum_{u'' \in N_j(v'')} \varphi_1\left(\Psi_{u''}^{(l-1)}\right)$$

where $v''$ can be a node from either $G$ or $G'$, and $\mathbb{I}[a] = 1$ if an event $a$ is true, $\mathbb{I}[a] = 0$ otherwise. $\qquad \square$

### A.3.4 Proof of Corollary 4.4

Corollary 4.4 states that with the same conditions in Theorem 4.1, for each $l$, in the limit on the layer width, the graph representation $\nu^{(l)}(G)$ recovers the empirical mean embedding $\hat{\tau}_{\mathbb{P}_G}^{(l)}$ of the domain distribution $\mathbb{P}_G$ in the reproducing kernel Hilbert space induced by $K_\nu^{(l)}$, given.

$$\hat{\tau}_{\mathbb{P}_G} = \frac{1}{|V|} \sum_{v \in V} \tilde{\Psi}_v^{(l)}$$

where $\tilde{\Psi}_v^{(l)}$ is the feature map of $K_\nu^{(l)}$ with $\tilde{\Psi}_v^{(l)} = \tilde{\varsigma}_w \sum_{i=0}^{k} \alpha_i \sum_{u \in N_i(v)} \varphi_1\left(\tilde{\Psi}_u^{(l-1)}\right)$.

*Proof.* We derive the feature map of domain-level kernel $K_\nu^{(l)}$ as follows. Similarly, we consider the case with $\tilde{\varsigma}_b = 0$, we have

$$K_\nu^{(l)}(G, G') = \sum_{i,j=0}^{k} \alpha_i \alpha_j' \left( \frac{\tilde{\varsigma}_w^2}{|V| \cdot |V'|} \mathbf{1}^T \mathbf{P}^{(i)} \mathbf{C}^{(l-1)}(\tilde{\varsigma}_w^2, \tilde{\varsigma}_b^2) \left(\mathbf{P}'^{(j)}\right)^T \mathbf{1} \right)$$

$$= \left\langle \frac{\tilde{\varsigma}_w}{|V|} \sum_{v \in V} \sum_{i=0}^{k} \alpha_i \sum_{u \in N_i(v)} \varphi_1\left(\tilde{\Psi}_u^{(l-1)}\right), \frac{\tilde{\varsigma}_w}{|V'|} \sum_{v' \in V'} \sum_{j=0}^{k} \alpha_j' \sum_{u' \in N_j(v')} \varphi_1\left(\tilde{\Psi}_{u'}^{(l-1)}\right) \right\rangle$$

That is, given the node representation $\tilde{\Psi}_v^{(l)} = \tilde{\varsigma}_w \sum_{i=0}^{k} \alpha_i \sum_{u \in N_i(v)} \varphi_1\left(\tilde{\Psi}_u^{(l-1)}\right)$ in the kernel space, the mean pooling in the READOUT function (see Eq.(2)) results in the mean embedding of empirical node distribution over $\{\tilde{\Psi}_v^{(l)}\}_{v \in V}$, i.e.,

$$\hat{\tau}_{\mathbb{P}_G} = \frac{1}{|V|} \sum_{v \in V} \tilde{\Psi}_v^{(l)}$$

Moreover, when the sample size of graph $G$ goes to infinity, the expected mean embedding of node distribution over $\{\tilde{\Psi}_v^{(l)}\}_{v \in V}$ is given by

$$\tau_{\mathbb{P}_G} = \mathbb{E}_v\left[\tilde{\Psi}_v^{(l)}\right] = \mathbb{E}_v\left[K_\nu^{(l)}(\cdot, v|G)\right] = \mathbb{E}_v\left[K_\nu^{(l)}(\cdot, v|\mathbb{P}_G)\right]$$

which corresponds to expression of graph domain distribution $\mathbb{P}_G$ in the reproducing kernel Hilbert space induced by $K_\nu^{(l)}$ (given by the kernel mapping $\tilde{\Psi}_v^{(l)}$). $\qquad \square$

### A.3.5 Proof of Theorem 4.5

Theorem 4.5 states that let $\mathbf{K}_{ss}^\mu$ be the sample-level covariance matrix over source nodes, i.e., $[\mathbf{K}_{ss}^\mu]_{(v_s, v_s')} = K_\mu^{(L)}(v_s, v_s'|G_s)$, $\nu_{ss} = K_\nu^{(L)}(G_s, G_s)$, $\nu_{tt} = K_\nu^{(L)}(G_t, G_t)$ be the intra-graph kernels and $\nu_{st} = K_\nu^{(L)}(G_s, G_t)$ be the inter-graph kernel. Suppose $\bar{\varrho}_s^2 \triangleq \left(\frac{\nu_{ss} \cdot \nu_{tt}}{\nu_{st} \cdot \nu_{st}} - 1\right) \sigma_w^2 \nu_{tt} \bar{\lambda}_{ss} + \frac{\nu_{tt} \cdot \nu_{tt}}{\nu_{st} \cdot \nu_{st}} \varrho_s^2$ where $\bar{\lambda}_{ss}$ is the maximum eigenvalue of $\mathbf{K}_{ss}^\mu$, for any $(v_t, G_t)$ the generalization error is bounded by

$$\epsilon_t \leq \int \Gamma_t(v_t, G_t; \bar{\varrho}_s^2, \varrho_t^2) p(v_t, G_t) d(v_t, G_t)$$

where $\Gamma_t(v_t, G_t; \bar{\varrho}_s^2, \varrho_t^2)$ is the variance assuming that all source examples are observed in the target domain with respect to noises $\bar{\varrho}_s^2, \varrho_t^2$.

*Proof.* The covariance matrix $\mathbf{K}$ over training samples can be rewritten as

$$\mathbf{K} = \begin{bmatrix} \mathbf{K}_{ss} & \mathbf{K}_{st} \\ \mathbf{K}_{ts} & \mathbf{K}_{tt} \end{bmatrix} = \begin{bmatrix} \sigma_w^2 \nu_{ss} \mathbf{K}_{ss}^\mu & \sigma_w^2 \nu_{st} \mathbf{K}_{st}^\mu \\ \sigma_w^2 \nu_{st} \mathbf{K}_{ts}^\mu & \sigma_w^2 \nu_{tt} \mathbf{K}_{tt}^\mu \end{bmatrix}$$
$$= \begin{bmatrix} \frac{\nu_{st}}{\nu_{tt}}\mathbf{I} & \mathbf{0} \\ \mathbf{0} & \mathbf{I} \end{bmatrix} \begin{bmatrix} \frac{\nu_{ss}\cdot\nu_{tt}}{\nu_{st}\cdot\nu_{st}}\sigma_w^2\nu_{tt}\mathbf{K}_{ss}^\mu & \sigma_w^2\nu_{tt}\mathbf{K}_{st}^\mu \\ \sigma_w^2\nu_{tt}\mathbf{K}_{ts}^\mu & \sigma_w^2\nu_{tt}\mathbf{K}_{tt}^\mu \end{bmatrix} \begin{bmatrix} \frac{\nu_{st}}{\nu_{tt}}\mathbf{I} & \mathbf{0} \\ \mathbf{0} & \mathbf{I} \end{bmatrix}$$

Then

$$\Gamma(v_t, G_t) = \mathbf{k}_{**} - \mathbf{k}_* \left(\mathbf{K}_{ss} + \varrho_s^2\mathbf{I}\right)^{-1} \mathbf{k}_*^T$$
$$= \mathbf{k}_{**} - \mathbf{k}_*^t \left(\Sigma\right)^{-1} \left(\mathbf{k}_*^t\right)^T$$

Following [9], we have

$$\Delta(v_t, G_t) = \mathbf{k}_*^t \left(\Sigma_t^{-1} - \Sigma^{-1}\right) \left(\mathbf{k}_*^t\right)^T$$

where

$$\Sigma = \begin{bmatrix} \frac{\nu_{ss}\cdot\nu_{tt}}{\nu_{st}\cdot\nu_{st}}\sigma_w^2\nu_{tt}\mathbf{K}_{ss}^\mu & \sigma_w^2\nu_{tt}\mathbf{K}_{st}^\mu \\ \sigma_w^2\nu_{tt}\mathbf{K}_{ts}^\mu & \sigma_w^2\nu_{tt}\mathbf{K}_{tt}^\mu \end{bmatrix} + \begin{bmatrix} \frac{\nu_{tt}\cdot\nu_{tt}}{\nu_{st}\cdot\nu_{st}}\varrho_s^2\mathbf{I} & \mathbf{0} \\ \mathbf{0} & \varrho_t^2\mathbf{I} \end{bmatrix}$$
$$\Sigma_t = \begin{bmatrix} \sigma_w^2\nu_{tt}\mathbf{K}_{ss}^\mu & \sigma_w^2\nu_{tt}\mathbf{K}_{st}^\mu \\ \sigma_w^2\nu_{tt}\mathbf{K}_{ts}^\mu & \sigma_w^2\nu_{tt}\mathbf{K}_{tt}^\mu \end{bmatrix} + \begin{bmatrix} s^2\mathbf{I} & \mathbf{0} \\ \mathbf{0} & \varrho_t^2\mathbf{I} \end{bmatrix}$$

Thus, it holds

$$\Delta(v_t, G_t) \leq 0$$
$$\Sigma_t^{-1} - \Sigma^{-1} \preceq 0$$
$$\Sigma_t \succeq \Sigma$$
$$\begin{bmatrix} \sigma_w^2\nu_{tt}\mathbf{K}_{ss}^\mu & \sigma_w^2\nu_{tt}\mathbf{K}_{st}^\mu \\ \sigma_w^2\nu_{tt}\mathbf{K}_{ts}^\mu & \sigma_w^2\nu_{tt}\mathbf{K}_{tt}^\mu \end{bmatrix} + \begin{bmatrix} s^2\mathbf{I} & \mathbf{0} \\ \mathbf{0} & \varrho_t^2\mathbf{I} \end{bmatrix} \succeq \begin{bmatrix} \frac{\nu_{ss}\cdot\nu_{tt}}{\nu_{st}\cdot\nu_{st}}\sigma_w^2\nu_{tt}\mathbf{K}_{ss}^\mu & \sigma_w^2\nu_{tt}\mathbf{K}_{st}^\mu \\ \sigma_w^2\nu_{tt}\mathbf{K}_{ts}^\mu & \sigma_w^2\nu_{tt}\mathbf{K}_{tt}^\mu \end{bmatrix} + \begin{bmatrix} \frac{\nu_{tt}\cdot\nu_{tt}}{\nu_{st}\cdot\nu_{st}}\varrho_s^2\mathbf{I} & \mathbf{0} \\ \mathbf{0} & \varrho_t^2\mathbf{I} \end{bmatrix}$$
$$\begin{bmatrix} \left(1 - \frac{\nu_{ss}\cdot\nu_{tt}}{\nu_{st}\cdot\nu_{st}}\right)\sigma_w^2\nu_{tt}\mathbf{K}_{ss}^\mu & \mathbf{0} \\ \mathbf{0} & \mathbf{0} \end{bmatrix} \succeq \begin{bmatrix} \left(\frac{\nu_{tt}\cdot\nu_{tt}}{\nu_{st}\cdot\nu_{st}}\varrho_s^2 - s^2\right)\mathbf{I} & \mathbf{0} \\ \mathbf{0} & \mathbf{0} \end{bmatrix}$$
$$\left(1 - \frac{\nu_{ss}\cdot\nu_{tt}}{\nu_{st}\cdot\nu_{st}}\right)\sigma_w^2\nu_{tt}\mathbf{K}_{ss}^\mu \succeq \left(\frac{\nu_{tt}\cdot\nu_{tt}}{\nu_{st}\cdot\nu_{st}}\varrho_s^2 - s^2\right)\mathbf{I}$$
$$\mathbf{K}_{ss}^\mu \preceq \frac{1}{\left(\frac{\nu_{ss}\cdot\nu_{tt}}{\nu_{st}\cdot\nu_{st}} - 1\right)\sigma_w^2\nu_{tt}}\left(s^2 - \frac{\nu_{tt}\cdot\nu_{tt}}{\nu_{st}\cdot\nu_{st}}\varrho_s^2\right)\mathbf{I}$$
$$\bar{\lambda}_{ss} \leq \frac{1}{\left(\frac{\nu_{ss}\cdot\nu_{tt}}{\nu_{st}\cdot\nu_{st}} - 1\right)\sigma_w^2\nu_{tt}}\left(s^2 - \frac{\nu_{tt}\cdot\nu_{tt}}{\nu_{st}\cdot\nu_{st}}\varrho_s^2\right)$$
$$s^2 \geq \left(\frac{\nu_{ss}\cdot\nu_{tt}}{\nu_{st}\cdot\nu_{st}} - 1\right)\sigma_w^2\nu_{tt}\bar{\lambda}_{ss} + \frac{\nu_{tt}\cdot\nu_{tt}}{\nu_{st}\cdot\nu_{st}}\varrho_s^2$$

Thus, the minimum of the upper bound is given by

$$\bar{\varrho}_s^2 \triangleq \left(\frac{\nu_{ss}\cdot\nu_{tt}}{\nu_{st}\cdot\nu_{st}} - 1\right)\sigma_w^2\nu_{tt}\bar{\lambda}_{ss} + \frac{\nu_{tt}\cdot\nu_{tt}}{\nu_{st}\cdot\nu_{st}}\varrho_s^2$$

$\square$

### A.3.6 Proof of Corollary 4.6

Corollary 4.6 states that when there are no labeled nodes in the target graph, i.e., $V_t^{la} = \emptyset$, it holds

$$\lim_{|V_s| \to \infty} \epsilon_t = \left(1 - \frac{\nu_{st}^2}{\nu_{tt}\nu_{ss}}\right) \int K^{(L)}\left((v_t, G_t), (v_t, G_t)\right) p(v_t, G_t)d(v_t, G_t)$$

where $\nu_{ss} = K_\nu^{(L)}(G_s, G_s), \nu_{tt} = K_\nu^{(L)}(G_t, G_t)$ are intra-graph kernels and $\nu_{st} = K_\nu^{(L)}(G_s, G_t)$ is inter-graph kernel.

*Proof.* It is given

$$K^{(l)}\left((v,G),(v',G')\right) = \sigma_w^2 \cdot K_\mu^{(l)}\left(v,v'|G,G'\right) \cdot K_\nu^{(l)}\left(G,G'\right)$$

For all $(v_t, G_t)$, it holds

$$\begin{aligned}
\Gamma(v_t, G_t) &= \mathbf{k}_{**} - \mathbf{k}_* \left(\mathbf{K}_{ss} + \varrho_s^2 \mathbf{I}\right)^{-1} \mathbf{k}_*^T \\
&= \sigma_w^2 \nu_{tt} K_\mu^{(l)}\left(v_t, v_t | G_t\right) \\
&\quad - \left(\sigma_w^2 \nu_{st} \mathbf{k}_{*\mu}\right) \left(\mathbf{K}_{ss} + \varrho_s^2 \mathbf{I}\right)^{-1} \left(\sigma_w^2 \nu_{st} \mathbf{k}_{*\mu}\right)^T \\
&= \sigma_w^2 \left(\nu_{tt} - \frac{\nu_{st}^2}{\nu_{ss}}\right) K_\mu^{(l)}\left(v_t, v_t | G_t\right) \\
&\quad + \frac{\nu_{st}^2}{\nu_{ss}^2} \left[\sigma_w^2 \nu_{ss} K_\mu^{(l)}\left(v_t, v_t | G_t\right) - \left(\sigma_w^2 \nu_{ss} \mathbf{k}_{*\mu}\right) \left(\mathbf{K}_{ss} + \varrho_s^2 \mathbf{I}\right)^{-1} \left(\sigma_w^2 \nu_{ss} \mathbf{k}_{*\mu}\right)^T\right] \\
&= \sigma_w^2 \left(\nu_{tt} - \frac{\nu_{st}^2}{\nu_{ss}}\right) K_\mu^{(l)}\left(v_t, v_t | G_t\right) + \frac{\nu_{st}^2}{\nu_{ss}^2} \Gamma_s(v_t, G_t)
\end{aligned}$$

where $\Gamma_s(v_t, G_t)$ is the posterior variance by considering $(v_t, G_t)$ as a virtual source sample. Then when the number of source samples goes to infinity, we have

$$\lim_{|V_s| \to \infty} \Gamma_s(v_t, G_t) = 0$$

Thus,

$$\begin{aligned}
\Gamma(v_t, G_t) &= \sigma_w^2 \left(\nu_{tt} - \frac{\nu_{st}^2}{\nu_{ss}}\right) K_\mu^{(l)}\left(v_t, v_t | G_t\right) \\
&= \left(1 - \frac{\nu_{st}^2}{\nu_{tt} \nu_{ss}}\right) \sigma_w^2 \nu_{tt} K_\mu^{(l)}\left(v_t, v_t | G_t\right) \\
&= \left(1 - \frac{\nu_{st}^2}{\nu_{tt} \nu_{ss}}\right) K^{(L)}\left((v_t, G_t),(v_t, G_t)\right)
\end{aligned}$$

where $\nu_{ss} = K_\nu^{(L)}(G_s, G_s), \nu_{tt} = K_\nu^{(L)}(G_t, G_t)$ denote the intra-graph kernels and $\nu_{st} = K_\nu^{(L)}(G_s, G_t)$ is the inter-graph kernel.

$$\begin{aligned}
\lim_{|V_s| \to \infty} \epsilon_t &= \int \Gamma(v_t, G_t) p(v_t, G_t) d(v_t, G_t) \\
&= \left(1 - \frac{\nu_{st}^2}{\nu_{tt} \nu_{ss}}\right) \int K^{(L)}\left((v_t, G_t),(v_t, G_t)\right) p(v_t, G_t) d(v_t, G_t)
\end{aligned}$$

The generalization error is determined by graph domain similarity $\frac{K_\nu^{(L)}(G_s, G_t)}{\sqrt{K_\nu^{(L)}(G_s, G_s) \cdot K_\nu^{(L)}(G_t, G_t)}}$. $\qquad \square$

## A.4 Model Discussion

In this section, we discuss several instantiations of the proposed structure-aware neural networks in Eq. (2) and their induced graph Gaussian processes. This shows the flexibility of the proposed algorithms in incorporating existing graph neural networks and graph kernels.

### A.4.1 Intantiations of Message-Passing Graph Neural Networks

Table 6 summarizes several instantiations of message-passing graph neural networks (GNNs) [18] of Eq. (1) with different neighborhood selection and message aggregation strategies.

It can be seen that homophily-based GNNs are more likely to aggregate messages from nearby neighbors, while heterophily-based GNNs have to adaptively aggregate messages from distant neighbors within the graph. These GNNs can be applied to design the proposed structure-aware neural networks in Eq. (2).

| Model | Message-passing mechanism | Neighborhood |
|---|---|---|
| GCN [25] | $h_v^{(l+1)} = \phi\left(\hat{A}_{vv}W^{(l)}h_v^{(l)} + \sum_{u \in N_1(v)} \hat{A}_{uv}W^{(l)}h_u^{(l)}\right)$ | $k = 1$ |
| GraphSAGE [20] | $h_v^{(l+1)} = \phi\left(W^{(l)}\left[h_v^{(l)}\|h_{N_1(v)}^{(l)}\right]\right)$ | $k = 1$ |
| GAT [50] | $h_v^{(l+1)} = \phi\left(\alpha_{vv}W^{(l)}h_v^{(l)} + \sum_{u \in N_1(v)} \alpha_{uv}W^{(l)}h_u^{(l)}\right)$ | $k = 1$ |
| GIN [59] | $h_v^{(l+1)} = \text{MLP}^{(l)}\left((1 + \epsilon^{(l)})h_v^{(l)} + \sum_{u \in N_1(v)} h_u^{(l)}\right)$ | $k = 1$ |
| GCNII* [10] | $h_v^{(l+1)} = \phi\left(\left((1 - \beta_l)\mathbb{I} + \beta_l W^{(l)}\right)\left((1 - \lambda_l)\sum_{u \in \{v\} \cup N_1(v)} \hat{A}_{uv}h_u^{(l)} + \lambda_l h_v^{(0)}\right)\right)$ | $k = 1$ |
| MixHop [1] | $h_v^{(l+1)} = \phi\left(W_0^{(l)}h_v\right)\|\phi\left(\sum_{u \in N_1(v)} \hat{A}_{uv}W_1^{(l)}h_u\right)\|\phi\left(\sum_{u \in N_2(v)} \hat{A}_{uv}^2 W_2^{(l)}h_u\right)$ | $k = 2$ |
| H$_2$GCN [68] | $h_v^{(l+1)} = \left(\sum_{u \in N_1(v)} \hat{d}_{uv}h_u^{(l)}\right) \| \left(\sum_{u \in N_2(v)} \hat{d}_{uv}h_u^{(l)}\right)$ | $k = 2$ |
| GPR-GNN [11] | $h_v^{(l)} = \sum_{i=1}^k \gamma_i \sum_{u \in V} \hat{A}_{uv}^i h_u^{(l-1)}$ | $k = 10$ |
| HOG-GCN [52] | $h_v^{(l+1)} = \phi\left(\mu W_e^{(l)}h_v^{(l)} + \xi W_n^{(l)} \sum_{u \in V}\left[\hat{D}^{-1}\left(A + A^2\right) \odot H\right]_{uv} h_u^{(l)}\right)$ | $k = 2$ |
| GloGNN* [29] | $h_v^{(l+1)} = (1 - \gamma)\sum_{u \in V} Z_{uv}^{(l)*}h_u^{(l)} + \gamma h_v^{(0)}$ | $k = \infty$ |

Table 6: Instantiations of message-passing graph neural networks (* initial residual connection is adopted)

### A.4.2 Instantiations of Structure-Aware Neural Networks

It can be seen that the definition of structure-aware neural network Eq. (2) is flexible to incorporate existing graph neural networks [10, 20, 25, 59] by instantiating $\mu^{(l)}(v|G)$ and $\nu^{(l)}(G)$. For example, we can use Graph Isomorphism Network (GIN) [25] to define the structure-aware neural network Eq. (2) as follows. The graph convolutional layer of GIN is

$$\mathbf{h}_v^{(l)} = \frac{1}{\sqrt{M}} \sum_{u \in \{v \cup N_1(v)\}} a_{uv}\mathbf{W}_{\text{GIN}}^{(l)}\mathbf{x}_u^{(l)} + \mathbf{b}_{\text{GIN}}^{(l)} \quad \text{and} \quad \mathbf{x}_u^{(l)} = \phi\left(\mathbf{h}_u^{(l-1)}\right) \tag{9}$$

where $a_{uv} = 1$ for $u \neq v$ and $a_{uv} = 1 + \epsilon^{(l)}$ for $u = v$. $\mathbf{W}_{\text{GIN}}^{(l)}$ and $\mathbf{b}_{\text{GIN}}^{(l)}$ denote the weight and bias parameters in GIN, respectively. In this case, we use GIN to instantiate both $\mu^{(l)}(v|G)$ and $\nu^{(l)}(G)$ for learning $\mathbf{h}_v^{(l)}$ and $\tilde{\mathbf{h}}_v^{(l)}$. In addition, the READOUT function of Eq. (2) can be instantiated with mean pooling [62], i.e., $\tilde{\mathbf{h}}_G^{(l)} = \frac{1}{|V|}\sum_{v \in V} \tilde{\mathbf{h}}_v^{(l)}$. It is revealed [21, 37] that GIN is equivalent to the Gaussian process in the limit on the layer width. Following this observation, the following theorem shows the equivalence between structure-aware neural network and adaptive graph Gaussian process when using GIN to instantiate Eq. (2).

**Theorem A.1.** *Suppose the graph convolutional layers of structure-aware neural network Eq. (2) are instantiated with GIN and the READOUT function of Eq. (2) is instantiated with mean pooling. Assume that the layer width the network width goes to infinity, and the model parameters of all neural layers are independently and randomly drawn from Gaussian distribution, then for each $i$ and $l$, the output function $f_i^{(l)}$ in Eq. (2) follows a Gaussian process with $f_i^{(l)} \sim \mathcal{GP}(0, K^{(l)})$, where the covariance function $K^{(l)}$ is given by*

$$K^{(l)}\left((v, G), (v', G')\right) = \sigma_w^2 \cdot K_\mu^{(l)}\left(v, v'|G, G'\right) \cdot K_\nu^{(l)}\left(G, G'\right) \tag{10}$$

*where*

$$K_\mu^{(l)}\left(v, v'|G, G'\right) = \varsigma_b^2 + \varsigma_w^2 \cdot [\mathbf{a}]_{v,:}\mathbf{C}^{(l-1)}[\mathbf{a}'^T]_{:,v'}$$

$$K_\nu^{(l)}\left(G, G'\right) = \varsigma_b^2 + \frac{\varsigma_w^2}{|V| \cdot |V'|}\mathbf{1}^T\mathbf{a}\mathbf{C}^{(l-1)}\mathbf{a}'^T\mathbf{1}$$

$$\mathbf{C}^{(l-1)} = \mathbb{E}_{z_i^{(l-1)} \sim \mathcal{GP}\left(0, K_\mu^{(l-1)}\right)}\left[\phi(z_i^{(l-1)})\phi(z_i^{(l-1)})^T\right]$$

*Here, $\mathbf{a}$ and $\mathbf{a}'^T$ are the adjacency matrices with $[\mathbf{a}]_{uv} = a_{uv}$ and $[\mathbf{a}']_{uv} = a'_{uv}$ respectively.*

*Proof.* The sample covariance is given by

$$K_\mu^{(l)}\left(v, v'|G, G'\right) = \mathbb{E}\left[\mu_j^{(l)}(v|G) \cdot \mu_j^{(l)}(v'|G')\right]$$

When learning the node representation, graphs $G = (V, E)$ and $G' = (V', E')$ can be considered as a single large graph $G'' = (V \cup V', E \cup E')$ with two disjoint components. The adjacent matrix of $G''$ is then given by $\mathbf{a}'' = \begin{bmatrix} \mathbf{a} & \mathbf{0} \\ \mathbf{0} & \mathbf{a}' \end{bmatrix}$. In this case, learning node representation of $v$ from $G$ is equivalent to learning that from $G''$, due to the disconnection of two components $G$ and $G'$ within the large graph $G''$. Then, using the results from [37], given a single graph $G''$, the outputs of each layer in GCN over all nodes are equivalent to the Gaussian process with zero mean and covariance matrix

$$K_\mu^{(l)}(V'', V'') = \varsigma_b^2 \mathbf{I} + \varsigma_w^2 \mathbf{a}'' \mathbf{C}_{G''}^{(l-1)} \mathbf{a}''^T$$

$$\mathbf{C}_{G''}^{(l-1)} = \mathbb{E}_{z_i^{(l-1)} \sim K_\mu^{(l-1)}(V'', V'')} \left[ \phi(z_i^{(l-1)}) \phi(z_i^{(l-1)})^T \right]$$

Considering the block matrix form of $\mathbf{C}_{G''}^{(l-1)} = \begin{bmatrix} \mathbf{C}_G^{(l-1)} & \mathbf{C}^{(l-1)} \\ \left(\mathbf{C}^{(l-1)}\right)^T & \mathbf{C}_{G''}^{(l-1)} \end{bmatrix}$ ($\mathbf{C}_{G''}^{(l-1)}$ is a symmetric matrix), we have

$$K_\mu^{(l)}(V'', V'') = \varsigma_b^2 \mathbf{I} + \varsigma_w^2 \begin{bmatrix} \mathbf{a} \mathbf{C}_G^{(l-1)} \mathbf{a}^T & \mathbf{a} \mathbf{C}^{(l-1)} \mathbf{a}'^T \\ \mathbf{a}' \left(\mathbf{C}^{(l-1)}\right)^T \mathbf{a}^T & \mathbf{a}' \mathbf{C}_{G'}^{(l-1)} \mathbf{a}'^T \end{bmatrix}$$

where

$$\mathbf{C}_G^{(l-1)} = \mathbb{E}_{z_i^{(l-1)} \sim K_\mu^{(l-1)}(V, V)} \left[ \phi(z_i^{(l-1)}) \phi(z_i^{(l-1)})^T \right]$$

$$\mathbf{C}^{(l-1)} = \mathbb{E}_{z_i^{(l-1)} \sim K_\mu^{(l-1)}(V, V')} \left[ \phi(z_i^{(l-1)}) \phi(z_i^{(l-1)})^T \right]$$

$$\mathbf{C}_{G'}^{(l-1)} = \mathbb{E}_{z_i^{(l-1)} \sim K_\mu^{(l-1)}(V', V')} \left[ \phi(z_i^{(l-1)}) \phi(z_i^{(l-1)})^T \right]$$

Therefore, it holds that

$$K_\mu^{(l)}(v, v'|G, G') = \varsigma_b^2 + \varsigma_w^2 \cdot [\mathbf{a} \mathbf{C}^{(l-1)} \mathbf{a}'^T]_{vv'} = \varsigma_b^2 + \varsigma_w^2 \cdot [\mathbf{a}]_{v,:} \mathbf{C}^{(l-1)} [\mathbf{a}'^T]_{:,v'}$$

where $[\mathbf{B}]_{vv'}$ denotes the entry of a matrix $\mathbf{B}$ at the $v$-th row and $v'$-th column.

In addition, the domain covariance is given by

$$K_\nu^{(l)}(G, G') = \mathbb{E} \left[ \frac{1}{|V| \cdot |V'|} \sum_{v \in V} \sum_{v' \in V'} \mu_j^{(l)}(v|G) \cdot \mu_j^{(l)}(v'|G') \right]$$

$$= \frac{1}{|V| \cdot |V'|} \sum_{v \in V} \sum_{v' \in V'} \mathbb{E} \left[ \mu_j^{(l)}(v|G) \cdot \mu_j^{(l)}(v'|G') \right]$$

$$= \frac{1}{|V| \cdot |V'|} \sum_{v \in V} \sum_{v' \in V'} \left( \varsigma_b^2 + \varsigma_w^2 \cdot [\mathbf{a}]_{v,:} \mathbf{C}^{(l-1)} [\mathbf{a}'^T]_{:,v'} \right)$$

$$= \varsigma_b^2 + \frac{\varsigma_w^2}{|V| \cdot |V'|} \mathbf{1}^T \mathbf{a} \mathbf{C}^{(l-1)} \mathbf{a}'^T \mathbf{1}$$

which completes the proof. $\qquad \square$

Similarly, we can also design the adaptive graph Gaussian processes for other message-passing graph neural networks. More generally, we have the following observations.

**Corollary A.2.** *Suppose that the layer width goes to infinity, and the model parameters are independently and randomly drawn from Gaussian distributions. If $\mu_j^{(l)}(v|G)$ and $\nu_j^{(l)}(G)$ are Gaussian processes, i.e., $\mu_j^{(l)}(v|G) \sim \mathcal{GP}(0, K_\mu^{(l)})$ and $\nu_j^{(l)}(G) \sim \mathcal{GP}(0, K_\nu^{(l)})$, then, for each $i$ and $l$, the output function $f_i^{(l)}(v, G)$ in Eq. (2) is a Gaussian process with $f_i^{(l)}(v, G) \sim \mathcal{GP}(0, K^{(l)})$, where the covariance matrix $K^{(l)}$ is given by*

$$K^{(l)}((v, G), (v', G')) = \sigma_w^2 \cdot K_\mu^{(l)}(v, v'|G, G') \cdot K_\nu^{(l)}(G, G')$$

*where $v \in V$ and $v' \in V'$ denote nodes within the graphs $G$ and $G'$, respectively.*

*Proof.* Since both $\mu_j^{(l)}(v|G)$ and $\nu_j^{(l)}(G)$ are Gaussian processes and the weight and bias parameters are independent and identically distributed (IID), the output $f_i^{(l)}(v, G)$ at the $l^{\text{th}}$ layer can be considered as a sum of IID terms. Based on the Central Limit Theorem, in the limit of infinite network width $N_l \to \infty$, $f_i^{(l)}(v, G)$ is Gaussian distributed. Moreover, using the multidimensional Central Limit Theorem, any finite collections $\{f_i^{(l)}(v_1, G), f_i^{(l)}(v_2, G), \cdots, f_i^{(l)}(v_m, G)\}$ have a joint multivariate Gaussian distribution. Therefore, we conclude that $f_i^{(l)}(v, G)$ forms a Gaussian process, i.e., $f_i^{(l)}(v, G) \sim \mathcal{GP}(\omega^{(l)}, K^{(l)})$. The mean function $\omega^{(l)}$ is given by

$$\omega^{(l)}(v, G) = \mathbb{E}\left[f_i^{(l)}(v, G)\right] = \mathbb{E}\left[\frac{1}{\sqrt{N_l}} \sum_{j=1}^{N_l} \mathbf{W}_{ij}^{(l)} \cdot \mu_j^{(l)}(v|G) \cdot \nu_j^{(l)}(G)\right] = 0$$

and the covariance function is given by

$$K^{(l)}\left((v, G), (v', G')\right) = \sigma_w^2 \cdot \mathbb{E}\left[\mu_j^{(l)}(v|G) \cdot \mu_j^{(l)}(v'|G')\right] \cdot \mathbb{E}\left[\nu_j^{(l)}(G) \cdot \nu_j^{(l)}(G')\right]$$
$$= \sigma_w^2 \cdot K_\mu^{(l)}(v, v'|G, G') \cdot K_\nu^{(l)}(G, G')$$

Moreover, we see that for any $i > 0$, $f_i^{(l)}(v, G)$ can form a Gaussian process with identical mean and covariance functions. $\square$

**Remark.** Corollary A.2 shows that the transferable graph Gaussian process can be derived when both $\mu_j^{(l)}(v|G) \sim \mathcal{GP}(0, K_\mu^{(l)})$ and $\nu_j^{(l)}(G) \sim \mathcal{GP}(0, K_\nu^{(l)})$ are Gaussian processes. It is flexible to adopt different GNNs to define $\mu_j^{(l)}$ and $\nu_j^{(l)}$. Moreover, optimizing the selection of kernel spaces over $\mu_j^{(l)}$ and $\nu_j^{(l)}$ might lead to better transferable graph learning performance. We would like to leave it as our future work.

### A.5   Graph Gaussian Processes of Heterophilic GNNs

In Section 5, in order to tackle the knowledge transferability over heterophilic graphs, we compare the proposed `GraphGP` algorithm with LINKXGP, MixHopGP, and H2GCNGP derived from LINKX [30], MixHop [1], H2GCN [68] respectively. The graph convolutional layer of MixHop [1] is defined as

$$\mathbf{H}^{(l+1)} = \Big\|_{j \in P} \phi\left(\hat{\mathbf{A}}^j \mathbf{H}^{(l} \mathbf{W}_j^{(l)} + \mathbf{b}_j^{(l)}\right)$$

where $\|$ denotes column-wise concatenation. For example, Table 6 shows the MixHop with $P = \{0, 1, 2\}$, which corresponds to the neighbors from at most 2-order neighborhoods. The graph convolutional layer of MixHop [1] can also be rewritten as

$$\mathbf{h}_v^{(l)} = \frac{1}{\sqrt{M}} \Big\|_{j \in P} \left(\sum_{u \in \{v \cup N_1^j(v)\}} \hat{\mathbf{A}}_{uv}^j \mathbf{W}_{j,\text{MixHop}}^{(l)} \mathbf{x}_u^{(l)} + \mathbf{b}_{j,\text{MixHop}}^{(l)}\right) \quad \text{and} \quad \mathbf{x}_u^{(l)} = \phi\left(\mathbf{h}_u^{(l-1)}\right)$$

where $N_1^j(v)$ denotes the first-order neighborhood of node $v$ under adjacent matrix $\hat{\mathbf{A}}^j$. The following theorem shows the graph Gaussian process (termed as MixHopGP) derived from MixHop.

**Corollary A.3** (Graph Gaussian Process of MixHop)**.** *Suppose that the layer width goes to infinity, and the model parameters are independently and randomly drawn from Gaussian distributions. Let* $\mathbf{h}_v^{(l)}(i)$ *be the $i^{th}$ entry of the output vector $\mathbf{h}_v^{(l)}$. Then for any $i$, $\mathbf{h}_v^{(l)}(i)$ of MixHop over all nodes* $\{v_1, \cdots, v_{|V|}\} \in V$ *follows Gaussian process with $\mathbf{h}_v^{(l)}(i) \sim \mathcal{GP}(0, K_{\text{MixHop}}^{(l)})$ where the covariance function $K_{\text{MixHop}}^{(l)}$ is given by*

$$K_{\text{MixHop}}^{(l)}(v, v') = \sum_{j \in P} \left(\varsigma_b^2 + \varsigma_w^2 \cdot [\hat{\mathbf{A}}^j]_{v,:} \mathbf{C}^{(l-1)} \left[\left(\hat{\mathbf{A}}^j\right)^T\right]_{:,v'}\right)$$

*where $v, v' \in V$ are two nodes within graph $G = (V, E)$.*

*Proof.* It can be proven using a similar idea in [37]. □

Similarly, we can show the graph Gaussian processes (terms as LINKXGP and H2GCNGP) for LINKX [30] and H2GCN [68] as follows.

**Corollary A.4** (Graph Gaussian Process of LINKX). *Suppose that the layer width goes to infinity, and the model parameters are independently and randomly drawn from Gaussian distributions. Let* $\mathbf{h}_v^{(l)}(i)$ *be the* $i^{th}$ *entry of the output vector* $\mathbf{h}_v^{(l)}$. *Then for any* $i$, $\mathbf{h}_v^{(l)}(i)$ *of LINKX over all nodes* $\{v_1, \cdots, v_{|V|}\} \in V$ *follows Gaussian process with* $\mathbf{h}_v^{(l)}(i) \sim \mathcal{GP}(0, K_{\mathrm{LINKX}}^{(l)})$ *where the covariance function* $K_{\mathrm{LINKX}}^{(l)}$ *is given by*

$$K_{\mathrm{LINKX}}^{(l)}(v, v') = \mathbf{C_A} + \mathbf{C_X} + \varsigma_b^2 + \varsigma_w^2 \cdot (\mathbf{C_A} + \mathbf{C_X})$$

*where* $\mathbf{C_A}, \mathbf{C_X}$ *corresponds the NNGP kernel [26] induced by the node attributes and adjacent matrix respectively.*

*Proof.* Following [26], the MLPs over node attributes and adjacent matrix are equivalent to Gaussian processes $\mathbf{C_A}, \mathbf{C_X}$. Then the graph Gaussian process of LINKX over $\mathbf{C_A}, \mathbf{C_X}$ can be derived using $\mathbf{y} = \mathrm{MLP}_f(\mathbf{W}[\mathbf{h_A}||\mathbf{h_X}] + \mathbf{h_A} + \mathbf{h_X})$ where $\mathbf{h_A}, \mathbf{h_X}$ are adjacent and feature hidden representation respectively. □

**Corollary A.5** (Graph Gaussian Process of H2GCN). *Suppose that the layer width goes to infinity, and the model parameters are independently and randomly drawn from Gaussian distributions. Let* $\mathbf{h}_v^{(l)}(i)$ *be the* $i^{th}$ *entry of the output vector* $\mathbf{h}_v^{(l)}$. *Then for any* $i$, *the output layer* $\mathbf{h}_v^{(L)}(i)$ *of H2GCN over all nodes* $\{v_1, \cdots, v_{|V|}\} \in V$ *follows Gaussian process with* $\mathbf{h}_v^{(L)}(i) \sim \mathcal{GP}(0, K_{\mathrm{H2GCN}}^{(L)})$ *where the covariance function* $K_{\mathrm{H2GCN}}^{(L)}$ *is given by*

$$K_{\mathrm{H2GCN}}^{(L)}(v, v') = \mathbf{C}^{(0)} + \sum_{l=1}^{L} \sum_{j=1}^{k} \left( \varsigma_b^2 + \varsigma_w^2 \cdot [\mathbf{A}_{\mathrm{H2GCN}}^j]_{v,:} \mathbf{C}^{(l-1)} \left[ \left(\mathbf{A}_{\mathrm{H2GCN}}^j\right)^T \right]_{:,v'} \right)$$

*where* $\mathbf{C}^{(0)}$ *is NNGP kernel [26] induced by ego-feature embedding layers.* $\mathbf{A}_{\mathrm{H2GCN}}^j$ *is the adjacent matrix of the* $j^{th}$ *neighborhood defined in H2GCN.*

*Proof.* The jumping knowledge is leveraged in designing the H2GCN for heterophilic graphs. Thus, we show the graph Gaussian process of H2GCN at the output layer. The output layer is given by $\mathbf{h}_v^{\mathrm{final}} = ||_{j \in \{0,1,\cdots,L\}} \mathbf{h}_v^{(l)}$. Following Corollary A.3, the Gaussian process of each layer can be derived and then combined in the output layer. □

### A.6 Efficiency Analysis

As illustrated in Subsection 4.3 and Algorithm 1, in the context of transferable graph learning, the hyperparameters of `GraphGP` can be optimized by maximizing the following log marginal likelihood

$$\log p(\mathbf{y}_{t_{la}}) = \log \left[ \mathcal{N} \left( \mathbf{y}_{t_{la}} | \mathbf{0}, \mathbf{K}_{t(s+t)} \mathbf{K}_{(s+t)(s+t)}^{-1} \mathbf{K}_{t(s+t)}^T + \varrho_t^2 \mathbf{I} \right) \right]$$

The crucial idea is to consider all the nodes $V_s \cup V_t^{la}$ as the inducing points [47].

However, the objective function involves the inversion of the covariance matrix with time complexity $\mathcal{O}((|V_s| + |V_t^{la}|)^3)$, thereby resulting in the intractable computation of `GraphGP` in large graphs. To this end, we provide a computationally efficient approximation in both model training and inference. Following [37, 47], we choose a subset of landmarks $V_{\mathrm{LM}}$ from the labeled source and target nodes. First, all the labeled target nodes are selected. Second, another $q \cdot |V_t^{la}|$ landmarks are randomly selected from the labeled source nodes. Thus, the landmarks $V_{\mathrm{LM}}$ have $(q+1) \cdot |V_t^{la}|$ nodes in total. In practice, a small value of $q$ leads to $(q+1) \cdot |V_t^{la}| \ll |V_s| + |V_t^{la}|$. Using the landmark nodes, the objective function of Eq. (7) can be approximated by

$$\log p(\mathbf{y}_{t_{la}}) = \log \left[ \mathcal{N} \left( \mathbf{y}_{t_{la}} | \mathbf{0}, \mathbf{K}_{t(\mathrm{LM})} \mathbf{K}_{(\mathrm{LM})(\mathrm{LM})}^{-1} \mathbf{K}_{t(\mathrm{LM})}^T + \varrho_t^2 \mathbf{I} \right) \right] \tag{11}$$

**Algorithm 1** `GraphGP`

---

**Input:** Source graph $G_s = (V_s, E_s)$ with labeled nodes $V_s$, target graph $G_t = (V_t, E_t)$ with labeled nodes $V_t^{la} \subset V_t$, number of neighborhoods $k$, layers of structure-aware neural network $L$.
**Output:** Predicted output values on target test nodes $V_* \subset V_t$.

1: Initialize the hyper-parameters of `GraphGP`;
2: **while** Stopping criterion is not satisfied **do**
3:     Calculate the covariance matrices $\mathbf{K}_{t(s+t)}, \mathbf{K}_{(s+t)(s+t)}$;
4:     Maximize the log marginal likelihood of Eq. (7);
5: **end while**
6: Estimate the posterior distribution of Eq. 6;
7: Output the prediction results $\boldsymbol{\gamma}(V_*, G_t)$.

---

where $\mathbf{K}_{(\mathrm{LM})(\mathrm{LM})}$ denotes the covariance matrix over landmarks, and $\mathbf{K}_{t(\mathrm{LM})}$ is the covariance matrix between labeled target nodes and landmarks. Following [37], the Nyström approximation of the posterior distribution in `GraphGP` can also be calculated over the selected landmarks. The proposed approximation strategy would reduce the overall time complexity from $\mathcal{O}((|V_s| + |V_t^{la}|)^3)$ to $\mathcal{O}(|V_s||V_t^{la}|^2)$.

## A.7 Additional Experiments

In this section, we provide more details and additional results for our experiments.

### A.7.1 Setup

**Data Sets:** We use the following graph learning benchmarks with node regression tasks.

- Twitch [44]: It has 6 different domains ("DE", "EN", "ES", "FR", "PT", and "RU"). In each domain, the graph represents the friendships among Twitch users, where each node corresponds to a Twitch user and each edge represents the mutual friendship of two users. Each node is also associated with an attribute vector encoding the game, location, and streaming habit information. Different from previous work [44] which considers the binary classification of whether a streamer uses explicit language, in this paper, we focus on the node regression task. Thus, we use the number of views in the original data[5] as the output value for each node/user.

- Agriculture [34, 60]: It has 3 different domains ("Maize" (MA), "Sorghum" (SG), and "Soybean" (SY)). The goal of agriculture analysis is to predict diverse traits (e.g., specific leaf area) of plants related to the plants' growth using leaf hyperspectral reflectance. The knowledge transferability across different species allows us to effectively infer the biochemical traits in real scenarios. Each data point represents a plant associated with a 1901-dimensional feature vector (e.g., spectral wavelengths 500-2400 nm). Following [55], we use $k$-NN to build the graph based on the feature similarity.

- Airports [43]: It has 3 different domains ("USA" (US), "Brazil" (BR), and "Europe" (EU)). Those domains involve the airport networks where each node corresponds to an airport and each edge represents the existence of commercial fights. Each node/airport is assigned a label based on their level of activity, measured in fights or people. Following [69], we use the degree-guided feature vector for each node.

- Wikipedia [44]: It has 3 different domains ("chameleon" (CH), "crocodile" (CR), and "squirrel" (SQ)). As introduced in [44], those domain data are given by Wikipedia page-page networks on different topics: chameleons, crocodiles, and squirrels. Each node corresponds to an article and each edge represents mutual links of two articles. The node feature is given by the presence of particular nouns in the articles and the average monthly traffic. It aims to predict the log average monthly traffic.

- WebKB [41]: It has 3 different domains ("Cornell" (CO), "Texas" (TX), and "Wisconsin" (WS)). It is collected from computer science departments of various universities by Carnegie

---

[5]`https://github.com/benedekrozemberczki/MUSAE`

Mellon University. Each node corresponds to a web page, and each edge represents the hyperlink of two web pages. The node features are given by the bag-of-words representation of web pages. The web pages are manually classified into five categories, student, project, course, staff, and faculty. In contrast to previous works [41], we would like to consider the regression task where each node is associated with an output value from $[0, 4]$, e.g., 4.0 for nodes from class 4.

The train/validation/test split in our experiments is given below. As discussed in Subsection 4.3, for transferable node regression, it is given a source graph $G_s = (V_s, E_s)$ with fully labeled nodes and a target graph $G_t = (V_t, E_t)$ with a limited number of labeled nodes (e.g., $|V_t^{la}| \ll |V_s|$ where $V_t^{la} \subset V_t$ is the set of labeled target nodes). Therefore, for Airport, Wikipedia, and WebKB data sets, we randomly select 10% of target nodes for the training set, 10% for the validation set, and 80% for the testing set. For Agriculture and Twitch data sets, we randomly select 1% of target nodes for the training set, 1% for the validation set, and 98% for the testing set.

In Subsection 5.1, we verify the positive correlation between transfer performance and graph domain similarity on Twitch (RU $\rightarrow$ PT). To this end, we gradually add some noise to the target graph such that the graph domain similarity between the source and target graphs can change accordingly. More specifically, given the feature vector (node attribute) $x_v$ of each node $v$, we simply multiply the feature vector by a constant $\delta_0$, i.e., $\hat{x}_v = \delta_0 \cdot x_v$, where $\delta_0$ denotes the perturbation magnitude. In the experiments, we choose $\delta_0$ from 0.1 to 1 with the interval of 0.1.

**Baselines:** We consider the following Gaussian process baselines. (1) RBFGP [42] and DINO [56] are feature-Only Gaussian processes without using graph structures. (2) GGP [35], SAGEGP [37], and GINGP [37] are graph Gaussian processes by considering source and target graphs as a large disjoint graph. (3) LINKXGP, MixHopGP, and H2GCNGP are graph Gaussian processes derived from LINKX [30], MixHop [1], H2GCN [68] respectively.

RBFGP [42] is a standard Gaussian process algorithm designed for IID data. DINO [56] considers the transfer learning scenarios by designing the adaptive transfer kernel between source and target domains. By incorporating the graph structures, GGP [35], SAGEGP [37], and GINGP [37] design the graph convolutional kernels. For example, recent work [37] shows the equivalence between graph neural networks (e.g., GraphSAGE [20], GIN [59]) and graph Gaussian processes. Similarly, we can derive the corresponding Gaussian processes for LINKX [30], MixHop [1] and H2GCN [68] in Appendix A.5, which are termed as LINKXGP, MixHopGP, and H2GCNGP, respectively. It is notable that for a fair comparison, all the baselines are trained over both labeled source and target nodes. For example, graph Gaussian processes can take the input source and target graphs as a large disjoint graph. In this case, the distribution shift between source and target graphs is not considered.

### A.7.2 Additional Results

**Learned Weight $\alpha_i$:** In Table 7, we report the learned weight $\alpha_i$ of `GraphGP` in the target domain for different data sets. For homophilic graphs, e.g., EU $\rightarrow$ BR of Airport, we see that the node itself and its first-order neighborhood have higher importance weights. In contrast, for heterophilic graphs, e.g., CO $\rightarrow$ TX of WebKB, the first-order neighborhood is less important for transferable graph learning. Furthermore, when source and target graphs follow different assumptions, e.g., TX (WebKB) $\rightarrow$ BR (Airport) or EU (Airport) $\rightarrow$ TX (WebKB), the learned weights on the target graph can also capture the homophily or heterophily properties of the target graph.

| Data | $\alpha_0$ | $\alpha_1$ | $\alpha_2$ |
|---|---|---|---|
| EU $\rightarrow$ BR | 0.9682 | 0.9118 | 0.3015 |
| CO $\rightarrow$ TX | 1.5201 | 0.0297 | 1.2054 |
| TX $\rightarrow$ BR | 1.1087 | 1.2848 | 0.1416 |
| EU $\rightarrow$ TX | 1.9503 | 0.0952 | 0.3609 |

Table 7: Neighborhood importance weight $\alpha_i$ in the target domain

**Impact of $L$:** We further investigate the impact of the number of layers $L$ in the proposed `GraphGP` algorithm. As shown in Table 8, `GraphGP` achieves relatively better results when $L \in \{2, 3, 4\}$. This observation is also consistent with the over-smoothing issues of conventional graph neural networks [20, 59].

| Model | $L=1$ | $L=2$ | $L=3$ | $L=4$ | $L=5$ |
|---|---|---|---|---|---|
| GraphGP | $0.7434_{\pm 0.0103}$ | $0.7909_{\pm 0.0382}$ | $0.8001_{\pm 0.0056}$ | $0.8028_{\pm 0.0075}$ | $0.7788_{\pm 0.0118}$ |

Table 8: Impact of $L$ on RU $\rightarrow$ PT of Twitch

**Efficiency Analysis:** We investigate the computational efficiency of GraphGP and its approximation GraphGP-E. Table 9 shows the performance of GraphGP-E on the Twitch data set[6]. It can be seen that GraphGP-E consistently outperforms GINGP on transferable graph learning. Moreover, Table 10 shows the effectiveness and efficiency of GraphGP-E with different values of $q$ on RU $\rightarrow$ PT of Twitch, where Full indicates the original GraphGP algorithm and the running time indicates the seconds per epoch. It can be seen that compared to GraphGP, GraphGP-E can achieve comparable performance and much higher computational efficiency.

| Model | DE $\rightarrow$ RU | DE $\rightarrow$ PT | DE $\rightarrow$ ES | DE $\rightarrow$ EN |
|---|---|---|---|---|
| GINGP [37] | $0.5159_{\pm 0.0047}$ | $0.5201_{\pm 0.0061}$ | $0.4861_{\pm 0.0070}$ | OOM |
| GraphGP-E | $0.7012_{\pm 0.0088}$ | $0.7351_{\pm 0.0427}$ | $0.7356_{\pm 0.0215}$ | $0.6916_{\pm 0.0057}$ |

Table 9: Performance of GraphGP-E with $q=10$ (OOM: Out of memory)

| Model | $q=1$ | $q=2$ | $q=3$ | $q=4$ | $q=5$ | $q=6$ | $q=7$ | $q=8$ | $q=9$ | $q=10$ | Full |
|---|---|---|---|---|---|---|---|---|---|---|---|
| $R^2$ | 0.4806 | 0.6064 | 0.6781 | 0.7315 | 0.7519 | 0.7495 | 0.7598 | 0.7626 | 0.7887 | 0.7960 | 0.7909 |
| Time (s) | 0.2169 | 0.2211 | 0.2281 | 0.2330 | 0.2406 | 0.2713 | 0.2763 | 0.2808 | 0.2871 | 0.2889 | 0.8761 |

Table 10: Efficiency

---

[6]The domain "FR" in Twitch is not used in the experiments, because it has some missing output values, i.e., the number of views.

