# OpenReview forum: "Graph-Structured Gaussian Processes for Transferable Graph Learning"
_NeurIPS.cc/2023/Conference — NeurIPS 2023 poster_

### Official Review · Reviewer_e15N · 2023-06-24

**Soundness:** 3 good
**Presentation:** 3 good
**Contribution:** 2 fair
**Rating:** 5
**Confidence:** 3

**Summary:**

This work proposes a generic graph-structured Gaussian process framework (GraphGP) to investigating the knowledge transferability between homophilic and heterophilic graphs. GraphGP uses a structure-aware neural network to encode local node representation and global graph representation (domain-level) simultaneously. In addition, a simple neighborhood selection strategy is designed to tackle the knowledge transferability in homophily and heterophily graphs. Experimental results demonstrate superior performance compared to baseline models.

**Strengths:**

(1) Concerning on the problem of transferable graph learning over non-IID graph data is interesting.

(2) The paper is clearly written and well formulated.

(3) This paper theoretically discussed the expressive power of GraphGP and analyze the knowledge transferability across graphs from the perspective of graph domain similarity.


**Weaknesses:**

(1) In the methodology section, the algorithm flow description of GraphGP should be clearer. It is recommended to add some brief descriptions of the algorithm flow, draw the algorithm table, and draw the overall framework figure.

(2) algorithm complexity analysis and convergence analysis should be described more specific.

(3) The theoretical analysis of the algorithm is sufficient but the experiment is weak. In order to verify the generalization of the model, more transfer learning tasks should be designed in the experiment, such as classification tasks.

(4) The experimental part should contain several real-world datasets to verify the effectiveness of the algorithm.


**Questions:**

(1) There are many hyperparameters in the GraphGP algorithm, and whether the performance of the model is sensitive to these hyperparameters?

(2) There are many network structures for graph representation learning in existing research. Why the author uses message-passing graph neural network to extract graph features? Can he be well applied in the task of transfer learning? Can it be well applied to transfer learning tasks?

(3) What are the limitations of the proposed methods?


**Limitations:**

Yes

---

> ### Author Rebuttal · Authors · 2023-08-10
>
> We would like to thank the reviewer for the insightful comments and valuable suggestions. Hereafter, we present responses addressing the concerns and queries raised by the reviewer.
>
> **Q1:** There are many hyperparameters in the GraphGP algorithm, and whether the performance of the model is sensitive to these hyperparameters?
>
> **A1:** As illustrated in lines 284-292, the hyperparameters involved in the proposed GraphGP algorithm were optimized by maximizing the log marginal likelihood in Eq. (7). In appendix A.7.2, we also reported the learned weights $\alpha_i$ after optimization. It is notable that hyperparameter optimization is commonly used in Gaussian process regression models [46]. The optimized hyperparameters tend to be positively correlated with the marginal likelihood, thus leading to better empirical performance in real scenarios.
>
> **Q2:** There are many network structures for graph representation learning in existing research. Why the author uses message-passing graph neural network to extract graph features? Can he be well applied in the task of transfer learning? Can it be well applied to transfer learning tasks?
>
> **A2:** Message-passing graph neural networks have achieved promising performance in a variety of graph mining tasks. They have also motivated recent transferable graph neural network models, e.g., GRADE [58], AdaGCN [14], EGI [71], etc. Thus, message-passing graph neural networks have been applied to transferable graph learning tasks. Following this direction, in this paper, we explored the theoretical understanding of message-passing graph neural networks on transferable graph learning tasks.
>
> **Q3:** What are the limitations of the proposed methods?
>
> **A3:** The limitations of the proposed methods are discussed in Appendix A.2. In this paper, we focus on the covariate shift assumption. In addition to covariate shift, label shift is another assumption in transfer learning scenarios. It can be challenging to extend the developed transferable graph Gaussian processes to tackle label shift scenarios. Besides, the proposed methods cannot directly handle the test-time adaptation scenarios where only testing samples are available in the target domain.
>
> **Q4:** In the methodology section, the algorithm flow description of GraphGP should be clearer. It is recommended to add some brief descriptions of the algorithm flow, draw the algorithm table, and draw the overall framework figure.
>
> **A4:** First of all, we would like to clarify that we have shown the algorithm details of GraphGP in Algorithm 1 in Appendix A.6. Second, following the reviewer's suggestion, we will provide more descriptions of the algorithm flow and the overall framework figure in the revised version.
>
> **Q5:** algorithm complexity analysis and convergence analysis should be described more specific.
>
> **A5:** We would like to clarify that in Appendix A.6, we did analyze the computational complexity of GraphGP. In addition, we also provided a more computationally efficient approximation of GraphGP and empirically evaluated this approximation approach in Tables 9 and 10.
>
> **Q6:** The theoretical analysis of the algorithm is sufficient but the experiment is weak. In order to verify the generalization of the model, more transfer learning tasks should be designed in the experiment, such as classification tasks.
>
> **A6:** Thank you for acknowledging our theoretical contributions on theoretically studying the knowledge transferability between source and target graphs, which is the main focus of this paper. Here we conduct additional experiments to adapt GraphGP to the classification task. Following previous work [41], we can use the one-hot representation of class labels to set up a multi-output regression problem. Table E reports the results (measured by node classification accuracy on the target graph) of GraphGP on node classification tasks. Here we use the social networks from [r3], where Blog 1 and Blog 2 are two disjoint social networks extracted from BlogCatalog. It further verifies the effectiveness of GraphGP over baselines on cross-network node classification tasks.
>
> | Model   | Blog 1 $\to$ Blog 2   | Blog 2 $\to$ Blog 1   |
> |---------|-----------------------|-----------------------|
> | RBFGP   | 0.2113$_{\pm 0.0087}$ | 0.2580$_{\pm 0.0083}$ |
> | GINGP   | 0.5190$_{\pm 0.0225}$ | 0.5097$_{\pm 0.0230}$ |
> | GraphGP | 0.5505$_{\pm 0.0052}$ | 0.5244$_{\pm 0.0162}$ |
> Table E: Performance comparison on cross-network node classification tasks
>
> [r3] Shen, Xiao, Quanyu Dai, Fu-lai Chung, Wei Lu, and Kup-Sze Choi. "Adversarial deep network embedding for cross-network node classification." In Proceedings of the AAAI conference on artificial intelligence, vol. 34, no. 03, pp. 2991-2999. 2020.
>
> **Q7:** The experimental part should contain several real-world datasets to verify the effectiveness of the algorithm.
>
> **A7:** We would like to clarify that we did use real-world data sets in our experiments. Take the agriculture data set as an example, it is collected by several universities for studying the relationship between diverse traits of plants related to the plant’s growth and the leaf hyperspectral reflectance. More details regarding the data description can be found in Appendix A.7.1.

---

> > ### Comment · Reviewer_e15N · 2023-08-16
> >
> > Thank the authors for providing a detailed rebuttal. Based on the contribution, I would like to keep the rating as borderline accept.

---

> > > ### Author Response · Authors · 2023-08-17
> > > **Thanks for Response**
> > >
> > > Dear Reviewer e15N,
> > >
> > > Thank you very much for your response.
> > >
> > >
> > > Best Regards,
> > >
> > > Authors of Submission13750

---

### Official Review · Reviewer_heNY · 2023-06-29

**Soundness:** 3 good
**Presentation:** 1 poor
**Contribution:** 2 fair
**Rating:** 4
**Confidence:** 3

**Summary:**

This paper deal with the ttransferable graph learning problem, especially between the homophily and heterophily graphs. To solve this problem, they propose a graph Gaussian process (GraphGP) algorithm, which is derived from a structure-aware neural network encoding both sample-level node representation and domain-level graph representation. The effectiveness of GraphGP is verified both theoretically and experimentally in various transferable node regression tasks.

**Strengths:**

1.	Comprehensive theoretical analysis.
2.	Clear problem definition.
3.	New techniques to consider the transfer learning between homophily and heterophily graphs.


**Weaknesses:**

1.	This paper has poor organization/presentation. There are two problem formulation and several proposed methods, not clear which is the major contribution. It is also not clear what is the relationship between the different mentioned techniques in this paper.
2.	Some comparison experiments are missing.


**Questions:**

1.	In introduction part, the author says “However, most existing works followed the IID assumption”. However, this seems not correct,  since graph data are not IID, and the existing transfer learning for graphs should also consider non-IID assumption. Could the author give more explanations on this?
2.	In line 52, the author mentions “structure-aware node”. It is better to provide the explanation of what kind of node can be named as “structure-aware”.
3.	In line 88, they author claims that “However, most existing works focus on either investigating the transferability of a pre-trained graph neural network”. It is not clear why the current existing transfer learning techniques for pretrained model cannot solve the problem formulated in this paper. This is a very important question to judge the contribution of the proposed techniques.
4.	In equation 2, what “i” and “j” refers to? And what is the meaning of “layer width”?
5.	In Section 4, the author introduces several methods, including “Structure-Aware Neural Network”. It is not clear if this method is proposed by author? If it is, the novelty is largely limited, since it is just the typical existing message passing network with a different theoretical analysis.
6.	The author formalized problems both in Section 3.1 and 4.3. It is confusing that which one is the goal of this paper.
7.	This paper only compares the proposed model with Gaussian models, while the existing graph transfer learning techniques are not compared.

---

> ### Author Rebuttal · Authors · 2023-08-10
>
> We greatly appreciate the reviewer's constructive comments. We would like to address the concerns and questions as follows.
>
> **Q1:** More explanations on “most existing works followed the IID assumption”?
>
> **A1:** In the introduction section, we started by introducing several transfer learning approaches, e.g., [3, 16, 39, 69, 8, 53, 24, 40, 65], under IID assumption that samples are independent and identically distributed in each domain (see lines 17-21). Knowledge transferability under IID assumption has been widely studied in past decades. In transfer learning scenarios, IID data (e.g., images for object recognition) are more frequently used compared to non-IID data. However, previous works under IID assumption [3, 16, 39, 69] might fail to handle transferability across domains with non-IID data. Furthermore, lines 46-50 discussed the existing transfer learning techniques for graphs under non-IID assumptions.
>
> **Q2:** Explanation of structure-aware nodes?
>
> **A2:** We would like to clarify that in this case, the "structure-aware input node" indicates the pair of inputs $(v, G)$ given a node $v$ and its associated graph $G$. Given a graph $G$ with $n$ nodes $\{v_1, v_2, \cdots, v_n\}$, the pairs of inputs can then be $\{(v_1, G), (v_2, G), \cdots, (v_n, G)\}$. Line 52 shows that we aim to leverage a structure-aware neural network to build the relationship between $(v_i, G)$ and $y_i$ for all $i=1,2\cdots,n$.
>
> **Q3:** Why existing TL techniques for pretrained model cannot solve the problem?
>
> **A3:** There are two major limitations in existing pre-trained graph neural networks. First, it did not leverage knowledge from source and target graphs to build a unified transfer learning framework. Intuitively, the crucial idea of our GraphGP algorithm is to find common knowledge (e.g., $S \cap T$) shared by source and target graphs via a unified transfer learning framework. In contrast, pre-trained GNNs might follow a two-stage framework: pre-training on source graph and then fine-tuning on target graph. That is, it first leverages the source knowledge (e.g., $S' \subset S$) to build GNN, and then investigates what knowledge (e.g., $S' \cap T$) in the learned source model is shared in the target graph. Thus, the two-stage transfer learning framework might lose some common knowledge compared to our unified framework, i.e., $S' \cap T \subset S \cap T$. Second, there is little theoretical analysis regarding the knowledge transferability induced by pre-trained graph neural networks. Compared to previous works [29, 49, 71], our paper theoretically shows the connection between graph domain similarity and knowledge transferability across domains (see Corollary 4.6 and Figure 3).
>
> **Q4:** In Eq. 2, what “i” and “j” refers to? What is “layer width”?
>
> **A4:** (a) In Eq. (2), $\mu^{(l)}(v|G)$ indicates the feature vector of node $v$ given graph $G$ at the $l$-th layer. Thus, $j$ in $\mu^{(l)}\_j(v|G)$ indicates the $j$-th dimension of node (sample-level) representation vector $\mu^{(l)}(v|G)$. Similarly, $j$ in $\nu^{(l)}\_j(G)$ indicates the $j$-th dimension of graph (domain-level) representation vector $\nu^{(l)}(G)$. $i$ in $f_i^{(l)}(v, G)$ indicates the $i$-th dimension of output feature vector $f^{(l)}(v, G)$ at the $l$-th layer. Furthermore, $\mathbf{W}_{ij}^{(l)}$ indicates the $i$-th row and $j$-th column of weight matrix $\mathbf{W}^{(l)}$ at the $l$-th layer.
>
> (b) Similar to previous work [41], "layer width" indicates the number of neurons in a graph convolutional layer.
>
> **Q5:** The novelty of the methods, e.g., “Structure-Aware Neural Network” in Section 4?
>
> **A5:** As illustrated in Subsection 4.1, we start by proposing the generic structure-aware neural network (see Eq. (2)). The major novelties of this proposed structure-aware neural network are two-fold. First, it incorporates both node (sample-level) representation and graph (domain-level) representation, in order to model source and target graphs for transferable graph learning. Second, as explained in lines 166-167, it is flexible to instantiate this generic structure-aware neural network with any message-passing GNN.
>
> Specifically, in Eqs. (3)(4), we provide simple instantiations of the structure-aware neural network of Eq. (2). As discussed in lines 176-177, similar message-passing strategies have also been considered in previous works [11, 56, 70]. Instead of developing novel message-passing strategies, this paper focuses on building the connections between the structure-aware neural network and the Gaussian process for transferable graph learning in Subsection 4.2. This connection enables us to develop the GraphGP algorithm in Subsection 4.3 and theoretically understand knowledge transferability across graphs in Subsection 4.4.
>
> **Q6:** It is confusing which one is the goal of this paper.
>
> **A6:** We would like to clarify that we provided the generic problem definition of transferable graph learning in Subsection 3.1. The goal is to learn the prediction function on the target graph, using knowledge from the source graph. In Subsection 4.3, we provided more detailed input and output of transferable graph learning on node regression tasks. The goal is still to learn a prediction function (more specifically, node regression function in this case) on the target graph, using knowledge from the source graph. Therefore, the problem definition in Subsection 3.1 is more generic, while in Subsection 4.3, we consider an instantiation of this generic problem setting based on the node regression task. Both of them have the same goal, i.e., learning the prediction function on the target graph using knowledge from the source graph. We will provide more clarification regarding the two problem definitions in the revised version.
>
> **Q7:** The existing graph TL techniques are not compared.
>
> **A7:** We would like to clarify that we did compare our GraphGP algorithm with existing graph transfer learning baselines (e.g., GRADE and AdaGCN) in Table 5.

---

> > ### Author Response · Authors · 2023-08-17
> > **Gentle Reminder**
> >
> > Dear Reviewer heNY,
> >
> > We would like to thank you again for your constructive comments and questions on our paper. We have carefully provided our responses to your raised questions and concerns. Please feel free to let us know if you have any other questions or concerns regarding our paper. Thanks for your time and consideration.
> >
> >
> > Best Regards,
> >
> > Authors of Submission13750

---

> > > ### Comment · Area_Chair_yR9Q · 2023-08-18
> > > **Thanks for authors' rebuttal!**
> > >
> > > Reviewer heNY, did the authors address your concerns about paper presentation and experimental comparisons? Thanks.

---

> > > > ### Author Response · Authors · 2023-08-21
> > > > **Gentle Reminder**
> > > >
> > > > Dear Reviewer heNY,
> > > >
> > > > As the deadline for the author-reviewer discussion is approaching, we are wondering whether you have additional concerns regarding our paper. Your feedback and insights are invaluable to us, and we greatly appreciate the time and effort you have already invested in reviewing our paper. To facilitate a constructive and thorough discussion, we kindly request that you take a moment to review our rebuttals at your earliest convenience. If you have any questions or concerns regarding the paper, please do not hesitate to reach out.
> > > >
> > > > Once again, thank you for your valuable comments on our paper. We look forward to your feedback and hope to address any remaining concerns to the best of our abilities.
> > > >
> > > > Best Regards,
> > > >
> > > > Authors of Submission13750

---

### Official Review · Reviewer_pydS · 2023-07-05

**Soundness:** 3 good
**Presentation:** 3 good
**Contribution:** 3 good
**Rating:** 6
**Confidence:** 3

**Summary:**

The paper studies transferable graph learning over non-IID graph data. In order to adapt the knowledge from source graphs to target graphs, the paper proposes a graph-structured Gaussian Process (GraphGP). The GraphGP is derived from a structure-aware neural network and due to the flexibility of the hyperparameters, GraphGP is able to transfer knowledge across different types of graphs, such as homophily graphs and heterophily graphs. Experimental results on five datasets show that the proposed GraphGP achieve better performance than Gaussian Process baselines.

**Strengths:**

1. The motivation of the proposed GraphGP is clear and the idea of using the Gaussian Process to address the graph transfer learning problem is novel and interesting.

2. Theoretical analyses are provided to justify the rationale of the proposed method.

3. Codes are provided for reproducibility.


**Weaknesses:**

1. It is not very clear from the paper how to implement the kernel function $K^{(L)}$ of GraphGP in the experiments. What kind of kernel function is used in the experiments?

2. The proposed GraphGP can only be used in the regression task. It seems that it is not easy to adapt the model to other graph learning tasks such as the classification task.

3. The paper only uses Gaussian Process models as baselines. Why not compare against some graph neural network models which are considered to be more effective in graph learning tasks?

4. The statistics of experimental datasets are missing. It’s unclear how the proposed GraphGP performs when dealing with graphs of different sizes.


**Questions:**

Please see the questions in the Weaknesses section.

**Limitations:**

The authors have discussed the limitations and broader impact of their work.

---

> ### Author Rebuttal · Authors · 2023-08-10
>
> Thank you very much for your thoughtful reviews and constructive questions about our paper. We appreciate the strengths you highlighted regarding our motivation and theoretical results on transferable graph learning. Here are our answers regarding the concerns.
>
> **Q1:** It is not very clear from the paper how to implement the kernel function $K^{L}$ of GraphGP in the experiments. What kind of kernel function is used in the experiments?
>
> **A1:** The calculation of $K^{(L)}$ is shown in Theorem 4.1. It can be seen that it follows an iterative computation process. More specifically, given base kernel $C^{(0)}$ over node attributes, both sample-level kernel $K^{(1)}\_{\mu}$ and domain-level kernel $K^{(1)}_{\nu}$ at the first layer can be computed based on $C^{(0)}$ and graph structure (i.e., neighbor selection). Thus, the kernel $K^{(1)}$ can be computed. These results can then be leveraged to compute $K^{(2)}$ at the second layer. It will continue this process until the final kernel $K^{(L)}$ is computed. In summary, the kernel function is iteratively computed based on the graph structure and the base kernel $C^{(0)}$ is defined over node attributes. As pointed out in [41], the base kernel $C^{(0)}$ can be any positive-definite kernel, e.g., linear kernel, RBF kernel, polynomial kernel, etc. Following [41], we adopted the RBF kernel for $C^{(0)}$ in the experiments.
>
> **Q2:** The proposed GraphGP can only be used in the regression task. It seems that it is not easy to adapt the model to other graph learning tasks such as the classification task.
>
> **A2:** We would like to point out that the proposed GraphGP can be adapted to handle classification tasks. There are two feasible solutions. (1) Following previous work [41], we can use the one-hot representation of class labels to set up a multi-output regression problem. In this case, each dimension (e.g., the $i$-th dimension of $\mathbf{y} \in R^{C}$) of output values of an input sample corresponds to whether this sample belongs to the $i$-th class (e.g., $\mathbf{y}_i \in \{0, 1\}$). (2) Another solution is to leverage the approximation techniques [r1, r2] to handle the non-Gaussian likelihood in the classification problem setting. For demonstration purposes, we conduct additional experiments to investigate the first solution by adapting GraphGP to the classification task. Table C reports the results (measured by node classification accuracy on the target graph) of GraphGP on node classification tasks. Here we use the social networks from [r3], where Blog 1 and Blog 2 are two disjoint social networks extracted from BlogCatalog. We would like to leave the graph-aware approximation solution for classification as our future work as it is beyond the scope of the current paper.
>
> | Model   | Blog 1 $\to$ Blog 2   | Blog 2 $\to$ Blog 1   |
> |---------|-----------------------|-----------------------|
> | RBFGP   | 0.2113$_{\pm 0.0087}$ | 0.2580$_{\pm 0.0083}$ |
> | GINGP   | 0.5190$_{\pm 0.0225}$ | 0.5097$_{\pm 0.0230}$ |
> | GraphGP | 0.5505$_{\pm 0.0052}$ | 0.5244$_{\pm 0.0162}$ |
> Table C: Performance comparison on cross-network node classification tasks
>
> [r1] Hensman, James, Alexander Matthews, and Zoubin Ghahramani. "Scalable variational Gaussian process classification." In Artificial Intelligence and Statistics, pp. 351-360. PMLR, 2015.
>
> [r2] Williams, Christopher KI, and Carl Edward Rasmussen. Gaussian processes for machine learning. Vol. 2, no. 3. Cambridge, MA: MIT press, 2006.
>
> [r3] Shen, Xiao, Quanyu Dai, Fu-lai Chung, Wei Lu, and Kup-Sze Choi. "Adversarial deep network embedding for cross-network node classification." In Proceedings of the AAAI conference on artificial intelligence, vol. 34, no. 03, pp. 2991-2999. 2020.
>
> **Q3:** The paper only uses Gaussian Process models as baselines. Why not compare against some graph neural network models which are considered to be more effective in graph learning tasks?
>
> **A3:** We would like to clarify that we did compare the proposed GraphGP method with recent transferable graph neural networks in Table 5. More specifically, we consider two recent transferable graph neural network models: GRADE [58] and AdaGCN [14]. Both of them design the transferable graph learning algorithm based on existing graph neural network architectures. The experimental results validated the effectiveness of our approach over these baselines.
>
> **Q4:** The statistics of experimental datasets are missing. It’s unclear how the proposed GraphGP performs when dealing with graphs of different sizes.
>
> **A4:** The data statistics are summarized as follows.
>
> | Data        |           | \# nodes | \# edges |
> |-------------|-----------|----------|----------|
> | Twitch      | DE        | 9,498    | 315,774  |
> |             | EN        | 7,126    | 77,774   |
> |             | ES        | 4,648    | 123,412  |
> |             | FR        | 6,551    | 231,883  |
> |             | PT        | 1,912    | 64,510   |
> |             | RU        | 4,385    | 78,993   |
> | Agriculture | Maize     | 182      | 364      |
> |             | Sorghum   | 1,610    | 3,220    |
> |             | Soybean   | 389      | 778      |
> | Airports    | USA       | 1,190    | 13,599   |
> |             | Brazil    | 131      | 1,038    |
> |             | Europe    | 399      | 5,995    |
> | Wikipedia   | Chameleon | 2,277    | 31,421   |
> |             | Crocodile | 11,631   | 170,918  |
> |             | Squirrel  | 5,201    | 198,493  |
> | WebKB       | Cornell   | 183      | 298      |
> |             | Texas     | 183      | 325      |
> |             | Wisconsin | 251      | 515      |
> Table D: Data statistics

---

> > ### Author Response · Authors · 2023-08-17
> > **Gentle Reminder**
> >
> > Dear Reviewer pydS,
> >
> > We would like to thank you again for your thoughtful reviews and constructive questions about our paper. We have carefully provided our answers to your raised questions and concerns. Please feel free to let us know if you have any other questions or concerns regarding our paper. Thanks for your time and consideration.
> >
> >
> > Best Regards,
> >
> > Authors of Submission13750

---

> > > ### Comment · Reviewer_pydS · 2023-08-17
> > >
> > > Thanks to the authors for providing detailed feedback regarding my questions about the paper. I would like to keep my score.

---

> > > > ### Author Response · Authors · 2023-08-21
> > > > **Thanks**
> > > >
> > > > Dear Reviewer pydS,
> > > >
> > > > Thank you very much for your comments.
> > > >
> > > > Best Regards,
> > > >
> > > > Authors of Submission13750

---

### Official Review · Reviewer_V7Dg · 2023-07-07

**Soundness:** 3 good
**Presentation:** 3 good
**Contribution:** 2 fair
**Rating:** 6
**Confidence:** 4

**Summary:**

This paper studies the problem of transferable graph learning involving knowledge transfer from a source graph to a relevant target graph. To solve this problem, the authors propose a graph Gaussian process (GraphGP) algorithm, which is derived from a structure-aware neural network encoding both sample-level node representation and domain-level graph representation. The efficacy of GraphGP is verified theoretically and empirically in various transferable node regression tasks.

**Strengths:**

1. This paper is well-organized and the presentation is good.
2. The authors propose a generic graph-structured Gaussian process framework, which  encodes local node representation (sample-level) and global graph representation (domain-level) simultaneously.
3. The proposed framework tackles the knowledge transferability in homophily and heterophily graphs using a simple neighborhood selection strategy.


**Weaknesses:**

1. Related work is inadequate. Graph transfer learning has also been studied in some important literature [1,2,3], but they are not discussed in this paper and should be adopted as baselines to compare with the proposed method.
2. The experimental part lacks the study of each component involved in the method, and the contribution of the proposed component to this paper cannot be proved, such as local node representation (sample-level), global graph representation (domain-level) and the neighborhood selection strategy.
3. The caption of Table 5 is too brief. It should contains more information to explain the table content.

[1] Zhu, Qi, et al. "Transfer learning of graph neural networks with ego-graph information maximization." Advances in Neural Information Processing Systems 34 (2021): 1766-1779.
[2] Han, Xueting, et al. "Adaptive transfer learning on graph neural networks." Proceedings of the 27th ACM SIGKDD Conference on Knowledge Discovery & Data Mining. 2021.
[3] Wu, Jun, Jingrui He, and Elizabeth Ainsworth. "Non-IID Transfer Learning on Graphs." Proceedings of the AAAI Conference on Artificial Intelligence. Vol. 37. No. 9. 2023.


**Questions:**

See above.

**Limitations:**

In addition to covariate shift, label shift is also commonly considered in transfer learning scenarios. It is much more challenging to extend the developed transferable graph Gaussian processes to tackle label shift scenarios.

---

> ### Author Rebuttal · Authors · 2023-08-10
>
> We would like to thank the reviewer for the constructive comments and suggestions. In the following, we present our responses addressing the raised concerns.
>
> **Q1:** Related work is inadequate. Graph transfer learning has also been studied in some important literature [1,2,3], but they are not discussed in this paper and should be adopted as baselines to compare with the proposed method.
>
> **A1:** We would like to clarify that in the paper, we did discuss the related work [1,3] (references [58, 71] in the original paper) and take [3] (reference [58] in the original paper) as one of our baselines in Table 5. In addition, we conduct additional experiments on Airport data sets. Table A further verifies the effectiveness of GraphGP compared to GRADE [3] and EGI [1]. Previous work [2] studied a different transferable graph learning problem. The major goal of [2] is to automatically select the most useful self-supervised tasks in the source graph to help the target task. Thus, it requires multiple unsupervised tasks within the source graph. In contrast, our work focused on modeling the transferability from the source graph with a single supervised task to the target graph. That is, [2] aimed to answer which source task should be leveraged to help the target task, whereas our work focused on answering how knowledge can be transferred across graphs with a single task.
>
> | Model   | BR $\to$ EU           | EU $\to$ BR           | BR $\to$ US           |
> |---------|-----------------------|-----------------------|-----------------------|
> | EGI [1]    | 0.5204$_{\pm 0.0357}$ | 0.4786$_{\pm 0.0225}$ | 0.4951$_{\pm 0.0176}$ |
> | GARDE [3]   | 0.5314$_{\pm 0.0208}$ | 0.4792$_{\pm 0.0296}$ | 0.4354$_{\pm 0.0109}$ |
> | GraphGP | 0.5567$_{\pm 0.0246}$ | 0.4983$_{\pm 0.0370}$ | 0.5293$_{\pm 0.0335}$ |
> Table A: Performance comparison between GraphGP and [1][3]
>
> **Q2:** The experimental part lacks the study of each component involved in the method, and the contribution of the proposed component to this paper cannot be proved, such as local node representation (sample-level), global graph representation (domain-level) and the neighborhood selection strategy.
>
> **A2:** In Table 7 of Appendix A.7.2, we reported the learned weight $\alpha_i$ of GraphGP in the target domain for different data sets. These results validated the effectiveness of our neighborhood selection strategy in capturing homophily information from local neighborhoods. Besides, we conduct additional ablation studies to validate the effectiveness of local node representation (sample-level), global graph representation (domain-level), and the neighborhood selection strategy. Table B shows the results on Airport and WebKB. Here we consider three variants of GraphGP. (1) GraphGP-Local: It is a variant of GraphGP with only local node representation (i.e., the global graph induced kernel $K_{\nu}$ is removed). (2) GraphGP-Global: Instead of using node representation in GraphGP, GraphGP-Global would directly use raw node attributes/features as local representation while global representation is learned as GraphGP. (3) GraphGP with $\alpha_i \equiv 1$: That is, GraphGP equally aggregates information from local neighborhoods. Table A shows that without local node presentation or global distribution information, the performance of GraphGP drops. Furthermore, without adaptively selecting neighbors, the potential heterophilic neighbors would significantly degrade the performance of GraphGP on WebKB.
>
> | Model                            | Airport                 |                         | WebKB                   |                         |
> |----------------------------------|-------------------------|-------------------------|-------------------------|-------------------------|
> |                                  | BR $\to$ EU             | BR $\to$ US             | CO $\to$ TX             | WS $\to$ TX             |
> | GraphGP-Local                    | 0.5475$_{\pm {0.0249}}$ | 0.4986$_{\pm {0.0273}}$ | 0.3998$_{\pm {0.0405}}$ | 0.3243$_{\pm {0.0506}}$ |
> | GraphGP-Global                   | 0.5229$_{\pm {0.0172}}$ | 0.4158$_{\pm {0.0220}}$ | 0.3344$_{\pm {0.0717}}$ | 0.3017$_{\pm {0.0314}}$ |
> | GraphGP with $\alpha_i \equiv 1$ | 0.5414$_{\pm {0.0127}}$ | 0.5118$_{\pm {0.0188}}$ | 0.1190$_{\pm {0.0298}}$ | 0.0632$_{\pm {0.0381}}$ |
> | GraphGP                          | 0.5567$_{\pm 0.0246}$   | 0.5293$_{\pm 0.0335}$   | 0.4146$_{\pm 0.0402}$   | 0.3301$_{\pm 0.0585}$   |
> Table B: Ablation studies on local node representation (sample-level), global graph representation (domain-level), and the neighborhood selection strategy
>
> **Q3:** The caption of Table 5 is too brief. It should contain more information to explain the table content.
>
> **A3:** As illustrated in lines 368-369,  Table 5 reports the results of the proposed GraphGP algorithm and transferable graph neural networks (GNNs) baselines on RU $\to$ PT of the Twitch data set. We will make the caption of Figure 5 much clearer in the revised version.

---

> > ### Comment · Reviewer_V7Dg · 2023-08-15
> >
> > Thanks for the authors' efforts for their response. My concerns have been well addressed and I would like to raise my score to 6.

---

> > > ### Author Response · Authors · 2023-08-16
> > > **Response by authors**
> > >
> > > Dear Reviewer V7Dg,
> > >
> > > Thank you very much for acknowledging that your concerns have been well addressed.
> > >
> > > Best Regards,
> > >
> > > Authors of Submission13750

---

### Decision · Program_Chairs · 2023-09-21

**Decision:**

Accept (poster)

**Comment:**

After discussion, most reviewers agree that the paper is well-written and most of the concerns are addressed after additional justification and experiments in the rebuttal. The negative reviewer does not respond to AC's query so AC takes a look at authors' response and ignores the rating.